# Bridging the Gap Between Homogeneous and Heterogeneous Asynchronous Optimization is Surprisingly Difficult

## Abstract

Modern large-scale machine learning tasks often require multiple workers, devices, CPUs, or GPUs to compute stochastic gradients in parallel and asynchronously to train model weights. Theoretical results typically distinguish between two settings: (i) the homogeneous setting, where all workers have access to the *same* data distribution, and (ii) the heterogeneous setting, where each worker operates on *different* data distributions. Known optimal time complexities in these settings reveal a significant gap, with far more pessimistic guarantees in the heterogeneous case. In this work, we investigate whether these pessimistic optimal time complexities can be overcome under different assumptions. Surprisingly, we show that improvement is provably impossible under widely used first- and second-order similarity assumptions for a broad family of algorithms. We then turn to the interpolation regime and demonstrate that the weak interpolation assumption alone is also insufficient. Finally, we introduce a minimal combination of irreducible assumptions, strong interpolation and the local Polyak-Łojasiewicz condition, to derive a new time complexity bound that matches the best-known result in the homogeneous setting, without requiring identical data distributions.

## 1 Introduction

We consider optimization problems described by

$$\min_{x \in \mathbb{R}^d} \left\{ f(x) := \tfrac{1}{n} \sum_{i=1}^{n} \mathbb{E}_{\xi_i \sim \mathcal{D}_i} \left[ f_i(x; \xi_i) \right] \right\}, \tag{1}$$

where $f_i : \mathbb{R}^d \times \mathbb{S}_{\xi_i} \to \mathbb{R}$ and $\xi_i$ is a random variable with distribution $\mathcal{D}_i$ on $\mathbb{S}_{\xi_i}$ for all $i \in [n]$. Let us denote $f_i(x) := \mathbb{E}_{\xi_i \sim \mathcal{D}_i} \left[ f_i(x; \xi_i) \right]$. In our setup, we have $n$ workers/clients/CPUs/GPUs working in parallel and asynchronously, and each worker $i$ has access only to the stochastic gradient $\nabla f_i(x; \xi_i)$ of the function $f_i$ for all $x \in \mathbb{R}^d$. We want to find a (possibly random) point $\bar{x}$ such that $\mathbb{E}[\|\bar{x} - x_*\|^2] \leq \varepsilon$, where $x_*$ is a solution of (1). Such a problem arises in many machine learning (ML), deep learning, federated learning (FL), and data science problems (Konečný et al., 2016; McMahan et al., 2017; Goodfellow et al., 2016).

We focus on the modern setup where many workers work together in a distributed environment, where the workers can have *arbitrarily computation behaviors* due to hardware delays or network connectivity problems. Most previous works typically assume that the workers have the same performance that does not change over time. In contrast, our focus is on the setting where the *computation times are heterogeneous* and non-constant.

In the literature, the optimization problem (1) in the asynchronous environment is considered in two regimes: i) *heterogeneous setting*, where the functions $f_i$ can be arbitrarily different; in the context of ML and FL, it means the workers have access to different datasets. ii) *homogeneous setting*, where the functions $f_i$ are equal; in the context of ML and FL, it means the workers have access to the same dataset (Koloskova et al., 2022; Mishchenko et al., 2022; Feyzmahdavian & Johansson, 2023).

### 1.1 Notations

$[n] := \{1, \ldots, n\}$; $\mathbb{N}_0 := \{0, 1, 2, \ldots\}$; $\|\cdot\|$ is the standard Euclidean norm; $\langle \cdot, \cdot \rangle$ is the standard dot product; $g = \mathcal{O}(f)$ : exist $C > 0$ such that $g(z) \leq C \times f(z)$ for all $z \in \mathcal{Z}$; $g = \Omega(f)$ : exist $C > 0$ such that $g(z) \geq C \times f(z)$ for all $z \in \mathcal{Z}$; $g = \Theta(f)$ : $g = \mathcal{O}(f)$ and $g = \Omega(f)$; $g = \widetilde{\Theta}(f)$ : the same as $g = \Theta(f)$ but up to logarithmic factors.

## 1.2 PREVIOUS WORK

**Oracle complexity.** In the classical optimization theory (Nemirovskij & Yudin, 1983), algorithms are compared in terms of *oracle calls*. Assume that the number of workers is one and we work with nonconvex functions and the following standard assumptions:

**Assumption 1.1** (Global smoothness)**.** The function $f$ is differentiable and $L$–smooth, i.e., $\|\nabla f(x) - \nabla f(y)\| \leq L \|x - y\|$ for all $x, y \in \mathbb{R}^d$.

**Assumption 1.2** (Unbiased and $\sigma^2$-variance-bounded noise)**.** For all $x \in \mathbb{R}^d$, stochastic gradients $\nabla f_i(x; \xi)$ are unbiased and $\sigma^2$-variance-bounded, i.e., $\mathbb{E}_{\xi_i}[\nabla f_i(x; \xi_i)] = \nabla f_i(x)$ and $\mathbb{E}_{\xi_i}[\|\nabla f_i(x; \xi_i) - \nabla f_i(x)\|^2] \leq \sigma^2$ for all $i \in [n]$, where $\sigma^2 \geq 0$.

It is well known (Arjevani et al., 2022; Carmon et al., 2020) that the optimal oracle complexity is $\mathcal{O}\left(L\Delta/\varepsilon + \sigma^2 L\Delta/\varepsilon^2\right)$ to find $\bar{x} \in \mathbb{R}^d$ such that $\mathbb{E}[\|\nabla f(\bar{x})\|^2] \leq \varepsilon$. It is attained by the vanilla SGD method: $x^{k+1} = x^k - \gamma \nabla f(x^k; \xi^k)$, where $\xi^k$ are i.i.d. random samples, $\Delta := f(x^0) - f^*$, $x^0 \in \mathbb{R}^d$ is a starting point, and $\gamma = \Theta\left(\min\{1/L, \varepsilon/L\sigma^2\}\right)$ is a step size. In the convex setting (Assumption 5.1), the optimal oracle complexity is $\Theta\left(\sqrt{L}R/\sqrt{\varepsilon} + \sigma^2 R^2/\varepsilon^2\right)$ (Lan, 2020; Nemirovskij & Yudin, 1983) to find $\bar{x} \in \mathbb{R}^d$ such that $\mathbb{E}[f(\bar{x})] - f(x_*) \leq \varepsilon$, where $R := \|x^0 - x_*\|$. In the $\mu$–strongly convex setting, the optimal complexity $\widetilde{\Theta}\left(\sqrt{L}/\sqrt{\mu} + \sigma^2/\mu^2\varepsilon^2\right)$ is to find $\bar{x} \in \mathbb{R}^d$ such that $\|\bar{x} - x_*\|^2 \leq \varepsilon$ (up to logarithmic factors).

**Oracle complexity with many workers.** Many works discovered oracle complexities with multiple workers. Arjevani & Shamir (2015); Scaman et al. (2017) analyze the heterogeneous convex setting and provide lower bounds when the workers are synchronized. Lu & De Sa (2021) consider the similar setup but in the nonconvex setting. Arjevani et al. (2020) analyze settings where methods receive delayed stochastic gradients. Woodworth et al. (2018) provide lower bounds for parallel setups with intermittent communications and delayed updates. The primary limitation of these results is the assumption that all workers have consistent computational performance, without accounting for individual delays, random lags, or variations in performance over time.

**Time complexity.** To address the problem of analyzing methods with workers having different computation capabilities and performances, Mishchenko et al. (2022) proposed to consider the *fixed computation model*. In this model, it is assumed that

worker $i$ requires at most $\tau_i$ seconds to calculate one stochastic gradient.

Without loss of generality, we assume that the times are sorted: $\tau_1 \leq \cdots \leq \tau_n$. One of the most popular methods is Asynchronous SGD (Lian et al., 2015; Zhang et al., 2015; Feyzmahdavian et al., 2016; Sra et al., 2016; Dutta et al., 2018; Stich & Karimireddy, 2020; Wu et al., 2022; Islamov et al., 2024). In the *homogeneous setting*, Mishchenko et al. (2022); Koloskova et al. (2022); Cohen et al. (2021) showed that Asynchronous SGD and Picky SGD can provably improve the performance of the synchronized Minibatch SGD method that does the steps $x^{k+1} = x^k - \gamma/n \sum_{i=1}^n \nabla f(x^k; \xi_i^k)$, where $\gamma$ is a stepsize, $\xi_i^k$ are i.i.d. samples, and $\nabla f(x^k; \xi_i^k)$ are calculated in parallel in $n$ workers. Minibatch SGD requires $\mathcal{O}\left(L\Delta/\varepsilon + \sigma^2 L\Delta/n\varepsilon^2\right)$ iterations (Cotter et al., 2011; Goyal et al., 2017; Gower et al., 2019) in the nonconvex setting. Moreover, Minibatch SGD converges after $\mathcal{O}\left(\max_{i \in [n]} \tau_i \times \left(L\Delta/\varepsilon + \sigma^2 L\Delta/n\varepsilon^2\right)\right)$ seconds because it waits for the slowest worker with $\max_{i \in [n]} \tau_i$ in every iteration. Asynchronous SGD, methods with the step $x^{k+1} = x^k - \gamma^k/n \sum_{i=1}^n \nabla f(x^{k-\delta_k}; \xi_i^{k-\delta_k})$ and $\delta_k$–delayed stochastic gradients, improve this time complexity to $\mathcal{O}((1/n \sum_{i=1}^n 1/\tau_i)^{-1} \left(L\Delta/\varepsilon + \sigma^2 L\Delta/n\varepsilon^2\right))$.

**Optimal time complexities in the heterogeneous and homogeneous settings.** Surprisingly, the time complexity can be further improved. In the nonconvex setup (under Assumptions 1.1, and 1.2), Tyurin & Richtárik (2023) formalized the notion of time complexities and showed that the *optimal*

*time complexity* is

$$T_{\text{homog}} := \Theta \left( \min_{m \in [n]} \left[ \left( \frac{1}{m} \sum_{i=1}^{m} \frac{1}{\tau_i} \right)^{-1} \left( \frac{L\Delta}{\varepsilon} + \frac{\sigma^2 L \Delta}{m \varepsilon^2} \right) \right] \right) \tag{2}$$

seconds *in the homogeneous setup* to find an $\varepsilon$–stationary point, achieved by the Rennala SGD method[1], where, without loss of generality, the times are sorted: $\tau_1 \leq \cdots \leq \tau_n$. *In the heterogeneous setup*, the optimal time complexity is

$$T_{\text{heter}} := \Theta \left( \tau_n \frac{L\Delta}{\varepsilon} + \left( \frac{1}{n} \sum_{i=1}^{n} \tau_i \right) \frac{\sigma^2 L \Delta}{n \varepsilon^2} \right), \tag{3}$$

achieved by the Malenia SGD method (we discuss the methods in detail in Section 2).

**Difference between the two settings.** Using the inequality of arithmetic and harmonic means, one can easily show that $T_{\text{homog}} \leq T_{\text{heter}}$ (ignoring constant factors). At the same time, the gap between the complexities can be arbitrarily huge. Indeed, when the performance $\tau_1$ of the fastest worker tends to 0, one can easily show that $T_{\text{homog}} \to 0$ and $T_{\text{heter}} \to \Theta \left( \tau_n L\Delta/\varepsilon + \left( \frac{1}{n} \sum_{i=2}^{n} \tau_i \right) \sigma^2 L\Delta/n\varepsilon^2 \right)$, and $T_{\text{heter}}$ improves by at most $\sum_{i=1}^{n} \tau_i / \sum_{i=2}^{n} \tau_i \leq 2$. While the improvement in the homogeneous setup is $\infty$. Consider another example when the performance $\tau_n$ of the slowest worker (straggler) tends to $\infty$. Then $T_{\text{heter}} \to \infty$ and $T_{\text{homog}} \to \Theta(\min_{m \in [n-1]}[(1/n \sum_{i=1}^{n} 1/\tau_i)^{-1} \left( L\Delta/\varepsilon + \sigma^2 L\Delta/m\varepsilon^2 \right)])$, so the complexity $T_{\text{homog}}$ is robust to stragglers unlike $T_{\text{heter}}$.

**Arbitrarily computation dynamics.** The previous discussion explain that a significant gap appears between homogeneous and heterogeneous problems under the fixed computation model. This "arithmetic mean vs harmonic mean gap" was also observed in (Tyurin, 2025), where the author generalizes the fixed computation model to the *universal computation model*, accounting for potential disruptions caused by hardware or network delays, and any variations in computation speeds. For simplicity, in this work, we will continue working with the fixed computation model, but we also show how our final results translate to the universal computation model in Section A.

**Convex world.** When we want to find a point $\bar{x}$ such that $\mathbb{E}\left[ f(\bar{x}) \right] - f^* \leq \varepsilon$ in the convex setup, the gap is similar. The optimal time complexity *in the homogeneous setup* is

$$\Theta \left( \min_{m \in [n]} \left[ \left( \frac{1}{m} \sum_{i=1}^{m} \frac{1}{\tau_i} \right)^{-1} \left( \frac{\sqrt{L}R}{\sqrt{\varepsilon}} + \frac{\sigma^2 R^2}{m \varepsilon^2} \right) \right] \right) \tag{4}$$

seconds (Tyurin & Richtárik, 2023). While the optimal time complexity *in the heterogeneous setup* is

$$\Theta \left( \tau_n \frac{\sqrt{L}R}{\sqrt{\varepsilon}} + \left( \frac{1}{n} \sum_{i=1}^{n} \tau_i \right) \frac{\sigma^2 R^2}{n \varepsilon^2} \right) \tag{5}$$

seconds under Assumptions 5.1, 1.1, and 1.2 (**our new contribution**, Theorem D.4; the final puzzle piece needed to reveal the systematic gap between the two settings). Both complexities are achieved by the accelerated versions of Rennala SGD and Malenia SGD accordingly.

**Strongly convex world.** Assume additionally that the function $f$ is $\mu$–strongly convex. Using reduction (Woodworth & Srebro, 2016), up to logarithmic factors, we can obtain the optimal time complexity

$$\widetilde{\Theta} \left( \min_{m \in [n]} \left[ \left( \frac{1}{m} \sum_{i=1}^{m} \frac{1}{\tau_i} \right)^{-1} \left( \sqrt{\frac{L}{\mu}} + \frac{\sigma^2}{m \varepsilon \mu} \right) \right] \right) \tag{6}$$

in the homogeneous setting and the optimal time complexity

$$\widetilde{\Theta} \left( \tau_n \sqrt{\frac{L}{\mu}} + \left( \frac{1}{n} \sum_{i=1}^{n} \tau_i \right) \frac{\sigma^2}{n \varepsilon \mu} \right) \tag{7}$$

in the heterogeneous setting when we want to find a point $\bar{x}$ such that $\mathbb{E}\left[ f(\bar{x}) \right] - f^* \leq \varepsilon$. Here we also observe a large gap between the settings. Note that the complexities (3), (5), and (7) can only be improved under additional assumptions because they are optimal.

---

[1] It can also be achieved by another recent optimal method, Ringmaster ASGD (Maranjyan et al., 2025)

> **Main question:** Having the systematic gap between the homogeneous and heterogeneous setups, the goal of this work is to identify theoretical assumptions that are as weak as possible to improve the results of asynchronous methods in heterogeneous scenarios. Under which assumptions can we improve the dependence on the arithmetic mean of $\{\tau_i\}$ (see (3), (5), and (7)) to the dependence on the harmonic mean of $\{\tau_i\}$ (see (2), (4), and (6))? Right now, the only possible way is to assume that the functions $\{f_i\}$ are equal—an assumption we clearly want to avoid in the heterogeneous setting. Is there any chance to relax this assumption?

Addressing the potential for improving the pessimistic guarantees in heterogeneous settings is a crucial endeavor for understanding parallel distributed methods.

### 1.3 CONTRIBUTIONS

We observe that both Rennala SGD and Malenia SGD can be unified under a more general framework, Weighted SGD, which provides a natural foundation for analyzing heterogeneous methods. Since breaking the lower bounds in the heterogeneous setting requires additional assumptions, we start by introducing as few as possible to determine when Weighted SGD can outperform Malenia SGD.

**Analysis of first- and second-order similarity.** First, we consider the celebrated *first- and second-order similarity* and, surprisingly, prove that even under these assumptions—no matter how close the functions $\{f_i\}$ are—Weighted SGD converges if and only if it again reduces to Malenia SGD. Thus, it is infeasible to break the dependence on the arithmetic mean of $\{\tau_i\}$ under these assumptions.

**Investigate the interpolation assumption.** Next, we decided to go in another direction and consider the *interpolation* assumption. Using Theorem 3.1, we demonstrate that operating in the *interpolation regime* is essential. Thus, we introduce two additional assumptions, strong interpolation and the local Polyak-Łojasiewicz condition, and prove that it is impossible to drop either of these assumptions for improvement.

**Bridging the gap.** By identifying this minimal set of assumptions, we derive a new time complexity result that matches the best-known bound in the *homogeneous* setting (Section 5.2), but without requiring the functions $f_i$ to be identical. Our theoretical results are validated numerically in Section H.

*To bridge the gap in Section 5.2, we need to introduce Assumptions 5.6 and 5.7. However, our primary goal was to illustrate and prove that these assumptions are indeed necessary. Merely stating the assumptions might not be convincing; this is why the central part of our paper investigates different assumptions and shows that most of them do not allow bridging the gap. We believe that the significance of our contribution lies in this exploration process. While previous work noted the existence of the gap, our contribution goes further by systematically investigating which assumptions are sufficient and which are insufficient to eliminate it.*

## 2 A UNIFYING PERSPECTIVE ON Rennala SGD AND Malenia SGD

We start our work by looking closer to the Rennala SGD and Malenia SGD methods (see Algorithm 1) that achieve the optimal time complexities (2) and (3) in the homogeneous and heterogeneous setting, accordingly. We now recall how they work. In every iteration, Rennala SGD and Malenia SGD ask all workers to calculate stochastic gradients asynchronously at **the same iterate** $x^k$. Assume that worker $i$ has calculated $B_i^k$ stochastic gradients for all $i \in [n]$ at the iteration $k$. Then the methods do the steps

$$x^{k+1} = x^k - \gamma g_{\mathsf{R}}^k, \quad g_{\mathsf{R}}^k := \frac{1}{\sum_{i=1}^n B_i^k} \sum_{i=1}^n \sum_{j=1}^{B_i^k} \nabla f_i(x^k; \xi_{ij}^k) \qquad \text{(Rennala SGD)}$$

and

$$x^{k+1} = x^k - \gamma g_{\mathsf{M}}^k, \quad g_{\mathsf{M}}^k := \frac{1}{n} \sum_{i=1}^n \frac{1}{B_i^k} \sum_{j=1}^{B_i^k} \nabla f_i(x^k; \xi_{ij}^k), \qquad \text{(Malenia SGD)}$$

---

**Algorithm 1** Weighted SGD (reduces to Malenia SGD or Rennala SGD when $w_i^k$ are chosen as $w_i^k = 1/B_i^k$ or $w_i^k = n/\sum_{i=1}^n B_i^k$, respectively)

---

1: **Input:** point $x^0$, stepsize $\gamma$, parameter $S$,
   weights $\{w_i^k\}$
2: **for** $k = 0, 1, \dots, K - 1$ **do**
3:     Ask all workers to calculate stochastic gradients at $x^k$
4:     Init $g_i^k = 0$ and $B_i^k = 0$
5:     **while** $\left(\frac{1}{n}\sum_{i=1}^n (w_i^k)^2 B_i^k\right)^{-1} \leq \frac{S}{n}$ **do**
6:         Wait for the next worker $j$
7:         Update $B_j^k = B_j^k + 1$
8:         Receive a calculated stochastic gradient $\nabla f_j(x^k; \xi_{j,B_j^k}^k)$
9:         $g_j^k = g_j^k + \nabla f_j(x^k; \xi_{j,B_j^k}^k)$
10:        Ask this worker to calculate a stochastic gradient at $x^k$
11:     **end while**
12:     $g_w^k := \frac{1}{n}\sum_{i=1}^n w_i^k g_i^k = \frac{1}{n}\sum_{i=1}^n w_i^k \sum_{j=1}^{B_i^k} \nabla f_i(x^k; \xi_{ij}^k)$
13:     $x^{k+1} = x^k - \gamma g_w^k$
14:     Stop all the workers' calculations (or ignore the unfinished calculations in the subsequent iterations)
15: **end for**

---

accordingly. Rennala SGD and Malenia SGD ask all workers calculating stochastic gradients until $\frac{1}{n}\sum_{i=1}^n B_i^k > S/n$ and $\left(\frac{1}{n}\sum_{i=1}^n 1/B_i^k\right)^{-1} > S/n$ correspondingly, where $S$ is a parameter. Hence, both methods asynchronously collect and aggregate stochastic gradients to compute $g_\mathsf{R}^k$ and $g_\mathsf{M}^k$, and then perform a descent step. However, the way the methods aggregate is both different and important. It turns out the variance of the Rennala SGD's update is smaller. Indeed, one can easily show that

$$\mathbb{E}\left[\left\|g_\mathsf{R}^k - \mathbb{E}\left[g_\mathsf{R}^k\right]\right\|^2\right] \leq \frac{\sigma^2}{n}\left(\frac{1}{n}\sum_{i=1}^n B_i^k\right)^{-1} \text{ and } \mathbb{E}\left[\left\|g_\mathsf{M}^k - \mathbb{E}\left[g_\mathsf{M}^k\right]\right\|^2\right] \leq \frac{\sigma^2}{n}\left(\frac{n}{\sum_{i=1}^n \frac{1}{B_i^k}}\right)^{-1}.$$

Thus, the variance of Rennala SGD improves with the *arithmetic mean* of $B_i^k$, while the variance of Malenia SGD improves with the *harmonic mean* of $B_i^k$, which can be much smaller. Why wouldn't we use Rennala SGD in all scenarios if it is better? Because $g_\mathsf{R}^k$ is biased if $\{f_i\}$ are non-homogeneous. In general, $\mathbb{E}\left[g_\mathsf{R}^k\right] \neq \frac{1}{n}\sum_{i=1}^n f_i(x)$, while it is always true that $\mathbb{E}\left[g_\mathsf{M}^k\right] = \frac{1}{n}\sum_{i=1}^n f_i(x)$.

> **Takeaway 1:** The optimal methods calculate stochastic gradients at the last fixed point but employ different asynchronous aggregation strategies.

Taking into account Takeaway 1, it is reasonable to investigate their generalization, called Weighted SGD:

$$x^{k+1} = x^k - \gamma g_w^k, \quad g_w^k := \frac{1}{n}\sum_{i=1}^n w_i^k \sum_{j=1}^{B_i^k} \nabla f_i(x^k; \xi_{ij}^k), \qquad \text{(Weighted SGD)}$$

where the weights $\{w_i^k\}$ are free parameters. If we take $w_i^k = n/\sum_{i=1}^n B_i^k$ for all $i \in [n]$, we get Rennala SGD with small variance. If we take $w_i^k = 1/B_i^k$, we get Malenia SGD with high variance but with an unbiased estimator. The weights enable interpolation between the methods.

Further, we assume that the workers send the same number of stochastic gradients in each iteration, i.e., $B_i^k = B_i$ for all $i \in [n], k \geq 0$, and the weights also do not change, i.e., $w_i^k = w_i$ for all $i \in [n], k \geq 0$. We also assume that $\frac{1}{n}\sum_{i=1}^n w_i B_i = 1$. Otherwise, we can simply reparametrize and take $\gamma := \gamma/(\frac{1}{n}\sum_{i=1}^n w_i B_i)$.

## 3 NON-CONVERGENCE OF Weighted SGD WITH $w^k \neq 1/B^k$

Our goal now is to understand the possibility of decreasing the variance of Malenia SGD by choosing appropriate weights $\{w_i\}$ in the heterogeneous setting such that Weighted SGD converges. We start with the following pessimistic result.

**Theorem 3.1.** *Consider the* Weighted SGD *method with quadratic optimization problems, where* $f_i(x) : \mathbb{R} \to \mathbb{R}$ *such that* $f_i(x) = 0.5(x - a_i)^2$ *and* $a_i \in \mathbb{R}$ *for all* $i \in [n]$. *Assume that there is no noise in the stochastic gradients, which means* $\nabla f_i(x; \xi_i) = \nabla f_i(x)$ *deterministically for all* $\xi_i \in \mathbb{S}_{\xi_i}$, $i \in [n]$, *and* $x \in \mathbb{R}^d$, *Then* Weighted SGD *converges to the minimum only if* $w_i B_i = 1$ *for all* $i \in [n]$ *(*Malenia SGD*-like weighing); either it does not converge or it converges to* $\frac{1}{n} \sum_{j=1}^{n} w_j B_j a_j$ *instead of* $\frac{1}{n} \sum_{i=1}^{n} a_i$.

The theorem says that we can not naively apply Weighted SGD in solving (1) and ensure that we can find a point that is close to a solution in the heterogeneous setting unless we take $w_i = 1/B_i$ for all $i \in [n]$ (what we want to avoid).

> **Takeaway 2:** Even for simple quadratic problems without stochasticity, there is no hope of using any averaging other than Malenia SGD. Thus, in general, we must rely on Malenia SGD with the pessimistic dependence on the arithmetic mean of $\{\tau_i\}$.

## 4 FIRST-ORDER AND SECOND-ORDER SIMILARITY DON'T HELP

The main problem with the example from Theorem 3.1 is that it represents a worst-case scenario. Clearly, we have to introduce *assumptions* to ensure that Weighted SGD converges with weights distinct from those of Malenia SGD, due to Theorem 3.1 and the fact that Malenia SGD is optimal. One of the most popular assumptions in the literature is *first-order and second-order similarity of the functions* (Arjevani & Shamir, 2015; Szlendak et al., 2021; Mishchenko et al., 2022):

**Assumption 4.1** (First-Order Similarity). The functions $f_i$ satisfy $\max_{i,j \in [n]} \|\nabla f_i(x) - \nabla f_j(x)\|^2 \leq \delta_1$ for all $x \in \mathbb{R}^d$ for some $\delta_1 \geq 0$. It implies $\frac{1}{n} \sum_{i=1}^{n} \|\nabla f_i(x) - \nabla f(x)\|^2 \leq \delta_1$ for all $x \in \mathbb{R}^d$.

**Assumption 4.2** (Second-Order Similarity). The functions $f_i$ satisfy $\max_{i,j \in [n]} \|\nabla^2 f_i(x) - \nabla^2 f_j(x)\|^2 \leq \delta_2$ for all $x \in \mathbb{R}^d$ for some $\delta_2 \geq 0$. It implies $\frac{1}{n} \sum_{i=1}^{n} \|\nabla^2 f_i(x) - \nabla^2 f(x)\|^2 \leq \delta_2$ for all $x \in \mathbb{R}^d$.

One might expect that when both $\delta_1$ or $\delta_2$ are small, it would be possible to exploit the similarity and design a method with smaller variance and better dependence on $\{\tau_i\}$. Surprisingly, it is not the case: for any $\delta_1 > 0$ or $\delta_2 \geq 0$, one can construct a problem for which only Malenia SGD converges:

**Theorem 4.3.** *Consider the* Weighted SGD *method with* $f_i(x) : \mathbb{R} \to \mathbb{R}$ *such that* $f_i(x) = \beta \langle a_i, x \rangle + \frac{\beta}{2} \|x\|^2$, $a_i \in \mathbb{R}$ *for all* $i \in [n]$, $\frac{1}{n} \sum_{i=1}^{n} a_i = 0$, *and* $\|a_i\| = 1$, *where* $\beta > 0$ *is free parameter. Assume that there is no noise in the stochastic gradients, which means* $\nabla f_i(x; \xi_i) = \nabla f_i(x)$ *deterministically for all* $\xi_i \in \mathbb{S}_{\xi_i}$, $i \in [n]$, *and* $x \in \mathbb{R}$, *Then* Weighted SGD *converges to the point* $\frac{1}{n} \sum_{j=1}^{n} w_j B_j a_j$ *instead of* $x_* = 0$, *Assumption 4.1 (the first-order similarity) is satisfied with* $\delta_1 = 2\beta^2$, *and Assumption 4.2 (the second-order similarity) is satisfied with* $\delta_2 = 0$.

*Remark* 4.4. Note that $\frac{1}{n} \sum_{j=1}^{n} w_j B_j a_j = x_* = 0$ for all $a_i \in \mathbb{R}$ if and only if $w_i B_i = 1$ for all $i \in [n]$ (Malenia SGD-like weighting).

Hence, for any small $\delta_1 > 0$ and $\delta_2 \geq 0$, convergence is only possible with Malenia SGD. Due to the construction in Theorems 4.3, we can choose any $\beta > 0$, and hence any $\delta_1 > 0$. No matter how close the functions are to each other, Weighted SGD can converge close to the solution only if $w_i B_i = 1$ for all $i \in [n]$. In view of this, we argue that additional assumptions about the first- and second-order similarity will not help to improve the time complexity of Malenia SGD.

*Remark* 4.5. For the construction in Theorem 4.3, we can also show that $\|\nabla f_i(x)\|^2 \leq 2 \|\nabla f(x)\|^2 + 2\beta^2$ for all $i \in [n]$, which corresponds to the $\rho$–*strong growth* condition when $\beta = 0$ and $\rho = 2$ (Schmidt & Roux, 2013). Since Theorem 4.3 holds for all $\beta > 0$, we have proved the result for a "slightly" broader class of problems and have "almost" established that, even under the *strong growth*

Table 1: The summary of our results and the time complexities (up to logarithmic factors) to get a point $\bar{x}_*$ such that $\mathbb{E}[\|\bar{x}_* - \bar{x}_*\|^2] \leq \varepsilon$ under the *fixed computation model* (worker $i$ requires at most $\tau_i$ seconds to calculate one stochastic gradient; $\tau_1 \leq \cdots \leq \tau_n$) and Assumptions 5.1, 5.2, 1.2, and 1.1, where $\bar{x}_*$ is the closest solution to $\bar{x}$. The table compares methods in the fully heterogeneous setting and lists the extra assumptions the methods require to work.

| Method | Time Complexity Guarantees (previous results) | Additional Assumptions |
|---|---|---|
| Minibatch SGD | $\tau_n \left( \frac{L}{\mu} + \frac{\sigma^2}{n\varepsilon\mu^2} \right)$ | — |
| Asynchronous SGD (Mishchenko et al., 2022) | $\left( \frac{1}{n} \sum_{i=1}^{n} \frac{1}{\tau_i} \right)^{-1} \left( \frac{L}{\mu} + \frac{\sigma^2}{n\varepsilon\mu^2} \right)$ | $\{f_i\}$ are equal $\mu$–strong convexity |
| Malenia SGD (Tyurin & Richtárik, 2023) (Theorem E.2) | $\tau_n \frac{L}{\mu} + \left( \frac{1}{n} \sum_{i=1}^{n} \tau_i \right) \frac{\sigma^2}{n\varepsilon\mu^2}$ | — |
| Rennala SGD (Tyurin & Richtárik, 2023) (Theorem E.1) | $\min_{m \in [n]} \left[ \left( \frac{1}{m} \sum_{i=1}^{m} \frac{1}{\tau_i} \right)^{-1} \left( \frac{L}{\mu} + \frac{\sigma^2}{m\varepsilon\mu^2} \right) \right]$ | $\{f_i\}$ are equal |
| **Lower Bounds (new results)** | | |
| Under the first-order and second-order similarity, the following results state that the family of methods Weighted SGD can converge if and only if it reduces to Malenia SGD: | | |
| Family of methods Weighted SGD (Theorem 4.3) | Only Malenia SGD converges | Assumptions 4.1 and 4.2 (first-order and second-order similarity don't help) |
| The following results state that the family of methods Weighted SGD (includes Rennala SGD and Malenia SGD) can not improve Malenia SGD for small $\varepsilon$ if we discard Assumption 5.6 or 5.7: | | |
| Family of methods Weighted SGD (Theorem 5.8) | $\geq \left( \frac{1}{n} \sum_{i=1}^{n} \tau_i \right) \frac{\sigma^2}{n\varepsilon\mu^2}$ | Assumptions 5.4 and 5.7 (weak interpolation is not enough) |
| Family of methods Weighted SGD (Theorem 5.9) | $\geq \left( \frac{1}{n} \sum_{i=1}^{n} \tau_i \right) \frac{\sigma^2}{n\varepsilon\mu^2}$ | Assumption 5.6 |
| **Upper Bound (new result)** | | |
| The following results state that under Assumption 5.6 or 5.7 it is possible to improve Malenia SGD: | | |
| Rennala SGD (Theorem 5.10) | $\min_{m \in [n]} \left[ \left( \frac{1}{m} \sum_{i=1}^{m} \frac{1}{\tau_i} \right)^{-1} \left( \frac{L_{\max}}{\mu} + \frac{\sigma^2}{m\varepsilon\mu^2} \right) \right]$ | Assumptions 5.6 and 5.7 (weaker than the equality of functions $\{f_i\}$) |

condition, only Malenia SGD converges to the minimum. Whether a similar result holds for the class of problems satisfying $\max_{i \in [n]} \|\nabla f_i(x)\|^2 \leq 2 \|\nabla f(x)\|^2$ for all $x \in \mathbb{R}^d$ remains an important open research question.

> **Takeaway 3:** Even with first-order and second-order similarity, when using the family of methods Weighted SGD, there is still no hope of using any averaging other than Malenia SGD.

## 5 UNDERSTANDING THE GAP VIA INTERPOLATION ASSUMPTIONS

To understand the problem, we now focus on the standard setting of convex smooth functions under the PŁ-condition, where the latter is a much weaker assumption than $\mu$–strong convexity (Karimi et al., 2016).

**Assumption 5.1** (Convexity). The functions $f_i$ are convex for all $i \in [n]$. The function $f$ attains a minimum at a (non-unique) point $x_* \in \mathbb{R}^d$.

**Assumption 5.2** (Global Polyak-Łojasiewicz condition). There exists $\mu > 0$ such that $\|\nabla f(x)\|^2 \geq 2\mu \left( f(x) - f^* \right)$ for all $x \in \mathbb{R}^d$, where $f^*$ is the finite optimal function value of $f$.

**Assumption 5.3** (Local smoothness). The functions $f_i$ are differentiable and $L_i$–smooth. We also define $L_{\max} := \max_{i \in [n]} L_i$. Note that $L \leq L_{\max}$.

Looking at Takeaways 2 and 3, we see that a different similarity assumption is required to close the gap between the heterogeneous and homogeneous results. Recall Theorem 3.1, which states that Weighted SGD converges to $1/n \sum_{j=1}^{n} w_j B_j a_j$ instead of $1/n \sum_{i=1}^{n} a_i$. These two expressions are equal only if the minima $a_i$ of the functions $f_i$ are the same. Therefore, to ensure convergence when $w_i B_i \neq 1$ and to understand the gap between the homogeneous and heterogeneous settings, Theorem 3.1 motivates us to explore an alternative assumption known as the *interpolation* assumption (Vaswani et al., 2019). This assumption provides another way to capture the similarity among the functions $f_i$ by requiring that they share the same set of minimizers as the function $f$.

**Assumption 5.4** (Weak Interpolation). If $x^*$ is a minimizer of $f$, that is, $\nabla f(x^*) = 0$, then $x^*$ is also a minimizer of each $f_i$ for all $i \in [n]$.

Interpolation is a property of the solutions of $f_i$, whereas the heterogeneity assumptions, Assumptions 4.1 and 4.2, concern the gradients and Hessians. These are different characteristics of $f_i$, and understanding their connection could be an important future work.

Under Assumption 5.4, Theorem 3.1 is not a barrier anymore. Assumption 5.4 is considered practical in modern optimization literature, as there is evidence that it holds for large deep learning models (Zou & Gu, 2019; Zhang et al., 2021). However, as we show next, this assumption alone is not sufficient to achieve improved time complexity, leading to yet another pessimistic result:

**Theorem 5.5.** *Consider the* Weighted SGD *method. Let us fix any* $\varepsilon, L_{\max}, R, \mu, \sigma^2 > 0$ *such that* $\mu < L_{\max}/(2n), \varepsilon < 0.01$, *and* $R > 10$. *For all* $B_1, \ldots, B_n \geq 0$ *and any possible choice of weights* $\{w_i(B_1, \ldots, B_n)\}$ *as functions of* $B_1, \ldots, B_n$, *there exist functions* $\{f_i\}$ *and stochastic gradients* $\{\nabla f_i(\cdot; \cdot)\}$ *such that* $\{f_i\}$ *satisfy Assumptions 5.1, 5.3, and 5.4, $f$ satisfies Assumptions 5.2 and 1.1 with $L = L_{\max}$, $\{\nabla f_i(\cdot; \cdot)\}$ satisfy Assumption 1.2 such that the method requires at least*

$$\Omega \left( \left( \frac{1}{n} \sum_{i=1}^{n} \tau_i \right) \frac{\sigma^2}{\varepsilon n \mu^2} \log \left( \frac{R^2}{\varepsilon} \right) \right)$$

*seconds to find $\varepsilon$–solution in terms of distances to the solution set, when the method starts at a point in a distance less or equal to $R$ to the closest solution.*

Thus, even under Assumption 5.4, we can not improve the arithmetic mean dependence on $\{\tau_i\}$.

> **Takeaway 4:** Using the weak interpolation assumption, which captures the similarity of the functions in a different way compared to first-order and second-order similarity, it is still infeasible to improve the pessimistic dependence on $\{\tau_i\}$ achieved by Malenia SGD using the family of methods Weighted SGD.

## 5.1 STRONG INTERPOLATION AND LOCAL PŁ CONDITION ARE BOTH REQUIRED

Once again, we need to go deeper and introduce additional assumptions to break the lower bound from Theorem 5.5. To further investigate the problem, we now turn to two related assumptions.

**Assumption 5.6** (Strong Interpolation). For all $i \in [n]$, a point $x^*$ is a minimizer of $f$, that is, $\nabla f(x^*) = 0$, *if and only if* it is also a minimizer of $f_i$.

This assumption is clearly stronger than the weak interpolation assumption since it requires all the functions to share the set of minimizers.

**Assumption 5.7** (Local Polyak-Łojasiewicz condition). There exists $\mu$ such that $\|\nabla f_i(x)\|^2 \geq 2\mu (f_i(x) - f_i^*)$ for all $x \in \mathbb{R}^d$ and for all $i \in [n]$, where $f_i^*$ is the finite optimal function value of $f_i$.

This assumption, unlike Assumption 5.2, requires each function to satisfy PŁ condition. It turns out again that if we do not assume *both* Assumption 5.6 and Assumption 5.7, then it is infeasible to get a time complexity faster than in Malenia SGD with any weights $\{w_i\}$ for $\varepsilon$ small enough. This statement is formalized in the following two theorems.

**Theorem 5.8.** *Consider the* Weighted SGD *method. Let us fix any* $\varepsilon, L_{\max}, R, \mu, \sigma^2 > 0$ *such that* $\mu < L_{\max}/(2n), \varepsilon < 0.01$, *and* $R > 10$. *For all* $B_1, \ldots, B_n \geq 0$ *and any possible choice of weights* $\{w_i(B_1, \ldots, B_n)\}$ *as functions of* $B_1, \ldots, B_n$, *there exist functions* $\{f_i\}$ *and stochastic gradients* $\{\nabla f_i(\cdot; \cdot)\}$ *such that* $\{f_i\}$ *satisfy Assumptions 5.1, 5.3, 5.4, and 5.7 (**do not satisfy Assumption 5.6 in general**), $f$ satisfies Assumptions 5.2 and 1.1 with $L = L_{\max}$, $\{\nabla f_i(\cdot; \cdot)\}$ satisfy Assumption 1.2 such that the method requires at least*

$$\Omega \left( \left( \frac{1}{n} \sum_{i=1}^{n} \tau_i \right) \frac{\sigma^2}{\varepsilon n \mu^2} \log \left( \frac{R^2}{\varepsilon} \right) \right)$$

*seconds to find $\varepsilon$–solution in terms of distances to the solution set, when the method starts at a point in a distance less or equal to $R$ to the closest solution.*

**Theorem 5.9.** *Consider the* Weighted SGD *method. Let us fix any* $\varepsilon, L_{\max}, R, \sigma^2 > 0$ *such that* $\mu < L_{\max}/(2n)$, $\varepsilon < 0.01$, *and* $R > 10$. *For all* $B_1, \ldots, B_n \geq 1$ *and any possible choice of weights* $\{w_i(B_1, \ldots, B_n)\}$ *as functions of* $B_1, \ldots, B_n$, *there exist functions* $\{f_i\}$ *and stochastic gradients* $\{\nabla f_i(\cdot; \cdot)\}$ *such that* $\{f_i\}$ *satisfy Assumptions* 5.1, 5.3, 5.4, *and* 5.6 *(**do not satisfy Assumption 5.7 in general**),* $f$ *satisfy Assumptions* 5.2 *and* 1.1 *with* $L = L_{\max}$, $\{\nabla f_i(\cdot; \cdot)\}$ *satisfy Assumption* 1.2, *such that the method requires at least*

$$\Omega\left(\left(\frac{1}{n}\sum_{i=1}^{n}\tau_i\right)\frac{\sigma^2}{\varepsilon n\mu^2}\log\left(\frac{R^2}{\varepsilon}\right)\right)$$

*seconds to find* $\varepsilon$*–solution in terms of distances to the solution set, when the method starts at a point in a distance less or equal to* $R$ *to the closest solution.*

> **Takeaway 5:** Even when the weak interpolation assumption is combined with only one of Assumptions 5.6 and 5.7, we still obtain only the arithmetic mean dependence on $\{\tau_i\}$.

Once we drop either Assumption 5.6 or Assumption 5.7, it becomes possible to construct a "bad" function (see the proof of Theorems 5.8 and 5.9) that provides no room for Weighted SGD to improve, regardless of the weight choices.

## 5.2 FINALLY BRIDGING THE GAP

However, if assume that both Assumption 5.6 and Assumption 5.7 hold, then, finally, we can proof the convergence with harmonic-like dependence on $\{\tau_i\}$ :

**Theorem 5.10.** *Let Assumptions* 5.1, 5.3, 1.2, 5.6, 5.7 *hold[2]. We choose* $w_i^k = n/\sum_{i=1}^{n} B_i^k$ *for all* $k \geq 0, i \in [n]$ *in Algorithm* 1 *(*Weighted SGD *reduces to* Rennala SGD*). We take* $\gamma = 1/L_{\max}$, $S = 4\sigma^2/\mu L_{\max}\varepsilon$, *and run* Rennala SGD *for* $k \geq \Omega\left(\frac{L_{\max}}{\mu}\log\frac{R^2}{\varepsilon}\right)$ *iterations, then* $\mathbb{E}\left[\left\|x^{k+1} - x_*^{k+1}\right\|^2\right] \leq \varepsilon$, *where* $x_*^{k+1}$ *is the closest solution to* $x^{k+1}$. *Moreover, under the fixed computation model, the method requires*

$$\mathcal{O}\left(\min_{m\in[n]}\left[\left(\frac{1}{m}\sum_{i=1}^{m}\frac{1}{\tau_i}\right)^{-1}\left(\frac{L_{\max}}{\mu} + \frac{\sigma^2}{m\varepsilon\mu^2}\right)\right]\log\frac{R^2}{\varepsilon}\right) \tag{8}$$

*seconds.*

Under weaker assumptions, without requiring the equality of the functions $\{f_i\}$, this theorem yields time complexity guarantees with a "harmonic"-like dependence on the times $\{\tau_i\}$ for the Rennala SGD method, improving upon the previous theoretical results in Theorem E.1 and (Tyurin & Richtárik, 2023).

Notice that the method in Theorem 5.10 is still biased because $\mathbb{E}\left[\sum_{i=1}^{n}\sum_{j=1}^{B_i^k}\nabla f_i(x; \xi_{ij}^k)/\sum_{i=1}^{n} B_i^k\right] \neq \nabla f(x)$ in general. That said, we can successfully prove the theorem under this constraint. One of the primary reasons for this is the right choice of *convergence metric*. Initially, we aimed to analyze the biased gradient estimator in terms of function values and gradient norms, trying to prove that the method returns a point $\bar{x}$ such that $\mathbb{E}[f(\bar{x})] - f^* \leq \varepsilon$ or $\mathbb{E}[\|\nabla f(\bar{x})\|^2] \leq \varepsilon$. However, the more appropriate approach is to show $\mathbb{E}[\|\bar{x} - x_*\|^2] \leq \varepsilon$. Using this convergence metric allows us to analyze the biased gradient estimator. This observation can be important on its own.

> **Takeaway 6:** Improving the pessimistic dependence in Malenia SGD is possible with Rennala SGD and the additional assumptions, Assumption 5.6 and Assumption 5.7, in convex optimization.

One interesting observation is that there is no single best method between Malenia SGD and Rennala SGD: either Malenia SGD is the fastest, or Rennala SGD is when both Assumption 5.6 and Assumption 5.7 hold. We could not find a regime in between where any other method or strategy would improve upon both Malenia SGD and Rennala SGD.

---

[2]It is well-know that Assumption 5.3 implies Assumption 1.1. In Section F, we prove that Assumptions 5.1, 5.6 and Assumption 5.7 with constant $\mu$ imply Assumption 5.2 with constant $\mu/4$.

## 6 CONCLUSION

In this work, we investigated various assumptions and setups in an effort to break the pessimistic dependence on $\{\tau_i\}$ achieved by Malenia SGD. We considered the first- and second-order similarity, strong growth, and interpolation assumptions. We proved that under the first- and second-order similarity assumptions, it is infeasible to improve the dependence on the arithmetic mean of $\{\tau_i\}$ within the family of Weighted SGD methods. We also showed that under weak interpolation (Assumption 5.4), it is likewise not possible to improve the result by Malenia SGD. Subsequently, we presented new theoretical results that provide improved time complexity guarantees in the heterogeneous setting, without assuming that the functions $f_i$ are identical (Theorem 5.10). These results are obtained under the standard assumptions of convex optimization, together with Assumptions 5.6 and 5.7.

Importantly, we have not merely introduced these assumptions to close the gap, but have shown that both Assumptions 5.6 and 5.7 are provably essential and non-relaxable for the general family of methods, underscoring the fundamental limits of what can be achieved in heterogeneous convex stochastic optimization.

There are many unexplored directions that can build on our initial results and observations. While we focused on the most common assumptions in federated and distributed learning, our findings may inspire the development of new assumptions and settings where it is possible to improve upon Malenia SGD. Moreover, our upper bounds and lower bounds were investigated in terms of $\mathbb{E}[\|x^k - x_*^k\|^2]$ convergence. It would be interesting to see whether similar results can be obtained in terms of $\mathbb{E}[\|\nabla f(x^k)\|^2]$ in the non-convex setting. While Weighted SGD is a natural abstraction of the two optimal methods, Rennala SGD and Malenia SGD, and is general enough for investigation, it would be interesting to analyze other classes of methods as well.

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

CONTENTS

# A

## A    ARBITRARILY COMPUTATION DYNAMICS

Our new result can be readily extended to *the universal computation model*. To encompass virtually all computation scenarios, assume that each worker $i$ performs computations based on a *computation power* function $v_i : \mathbb{R}_+ \to \mathbb{R}_+$. Then the number of stochastic gradients that worker $i$ can calculate from a time $t_0$ to a time $t_1$ is an integral of the computation power $v_i$ followed by the floor operation:

$$\text{"\# of stoch. grad. in } [t_0, t_1]\text{"} = \left\lfloor \int_{t_0}^{t_1} v_i(\tau)d\tau \right\rfloor. \tag{9}$$

For instance, if worker $i$ is inactive for the first $t$ seconds and then active again, it would mean $v_i(\tau) = 0$ for all $\tau \leq t$ and $v_i(\tau) > 0$ for all $\tau > t$. Using the universal computation model, we can prove the theorem:

**Theorem A.1.** *Consider the assumptions, algorithm, and parameters from Theorem 5.10. Then,* Rennala SGD *converges after at most* $\bar{t}_{\left\lceil c \times \frac{L_{\max}}{\mu} \log \frac{R^2}{\varepsilon} \right\rceil}$ *seconds, where the sequence* $\{\bar{t}_k\}$ *is defined recursively as* $\bar{t}_k :=$

$$\min \left\{ t \geq 0 : \sum_{i=1}^{n} \left\lfloor \int_{\bar{t}_{k-1}}^{t} v_i(\tau)d\tau \right\rfloor \geq \max \left\{ \left\lceil \frac{\sigma^2}{\varepsilon} \right\rceil, 1 \right\} \right\} \tag{10}$$

*for all $k \geq 1$ ($\bar{t}_0 \equiv 0$), and $c$ is a universal constant.*

The similar result was obtained in (Tyurin, 2025). However, Tyurin (2025) requires the equality of the functions $\{f_i\}$ to get the sequence (10).

## B    PROOF OF THE MAIN RESULTS

**Theorem 3.1.** *Consider the* Weighted SGD *method with quadratic optimization problems, where $f_i(x) : \mathbb{R} \to \mathbb{R}$ such that $f_i(x) = 0.5(x - a_i)^2$ and $a_i \in \mathbb{R}$ for all $i \in [n]$. Assume that there is no noise in the stochastic gradients, which means $\nabla f_i(x; \xi_i) = \nabla f_i(x)$ deterministically for all $\xi_i \in \mathbb{S}_{\xi_i}, i \in [n]$, and $x \in \mathbb{R}^d$, Then* Weighted SGD *converges to the minimum only if $w_i B_i = 1$ for all $i \in [n]$ (*Malenia SGD*-like weighing); either it does not converge or it converges to $\frac{1}{n}\sum_{j=1}^{n} w_j B_j a_j$ instead of $\frac{1}{n}\sum_{i=1}^{n} a_i$.*

*Proof.* If $w_i B_i = 1$ for all $i \in [n]$, then Weighted SGD converges with $\gamma < 2$ because Malenia SGD converges. If $w_i B_i \neq 1$ for all $i \in [n]$, then

$$x^{k+1} = x^k - \gamma \frac{1}{n} \sum_{i=1}^{n} w_i B_i (x^k - a_i)$$

$$= \left(1 - \gamma \frac{1}{n} \sum_{i=1}^{n} w_i B_i\right)^{k+1} x^0 + \sum_{j=0}^{k} \gamma \left(1 - \gamma \frac{1}{n} \sum_{i=1}^{n} w_i B_i\right)^j \frac{1}{n} \sum_{i=1}^{n} w_i B_i a_i$$

$$= (1 - \gamma)^{k+1} x^0 + \sum_{j=0}^{k} \gamma (1 - \gamma)^j \frac{1}{n} \sum_{i=1}^{n} w_i B_i a_i,$$

where the third equality due to the agreement $\frac{1}{n}\sum_{i=1}^{n} w_i B_i = 1$.

If $\gamma \geq 2$, then $x^{k+1}$ does not converge if $x^0 \neq \frac{1}{n}\sum_{i=1}^{n} w_i B_i a_i$ and $k \to \infty$. If $\gamma < 2$, then

$$\lim_{k \to \infty} x^{k+1} = \frac{1}{n}\sum_{j=1}^{n} w_j B_j a_j.$$

$\square$

*Remark* B.1. It is possible to find $\{a_i\}_{i=1}^n$, when $\frac{1}{n}\sum_{j=1}^n w_j B_j a_j$ does not equal $\frac{1}{n}\sum_{i=1}^n a_i$

*Proof.* If $w_i \neq 1/B_i$ for all $i \in [n]$, then there exists $k_1 \in [n]$ such that $\frac{1}{n} w_{k_1} B_{k_1} < \frac{1}{n}$. If we take $a_{k_1} = 2$ and $a_i = 1$ for all $i \notin k_1$, then

$$\frac{1}{n}\sum_{j=1}^n w_j B_j a_j = \left(\frac{2}{n} w_{k_1} B_{k_1} + \left(1 - \frac{1}{n} w_{k_1} B_{k_1}\right)\right) > \frac{1+n}{n} = \frac{1}{n}\sum_{i=1}^n a_i,$$

and the method converges to the point that it is not equal to $\frac{1}{n}\sum_{i=1}^n a_i$. □

**Theorem 5.10.** *Let Assumptions 5.1, 5.3, 1.2, 5.6, 5.7 hold[3]. We choose $w_i^k = n/\sum_{i=1}^n B_i^k$ for all $k \geq 0, i \in [n]$ in Algorithm 1 (Weighted SGD reduces to Rennala SGD). We take $\gamma = 1/L_{\max}$, $S = 4\sigma^2/\mu L_{\max}\varepsilon$, and run Rennala SGD for $k \geq \Omega\left(\frac{L_{\max}}{\mu}\log\frac{R^2}{\varepsilon}\right)$ iterations, then $\mathbb{E}\left[\left\|x^{k+1} - x_*^{k+1}\right\|^2\right] \leq \varepsilon$, where $x_*^{k+1}$ is the closest solution to $x^{k+1}$. Moreover, under the fixed computation model, the method requires*

$$\mathcal{O}\left(\min_{m \in [n]}\left[\left(\frac{1}{m}\sum_{i=1}^m \frac{1}{\tau_i}\right)^{-1}\left(\frac{L_{\max}}{\mu} + \frac{\sigma^2}{m\varepsilon\mu^2}\right)\right]\log\frac{R^2}{\varepsilon}\right) \tag{8}$$

*seconds.*

*Proof.* Let us define $x_*^k$ as an euclidean projection of the point $x^{k+1}$ on to the solution set of the main problem (1), and take the condition expectation $\mathbb{E}_k[\cdot]$ w.r.t. the randomness from the iteration $k$ only. Then we have

$$\mathbb{E}_k\left[\left\|x^{k+1} - x_*^{k+1}\right\|^2\right] \leq \mathbb{E}_k\left[\left\|x^{k+1} - x_*^k\right\|^2\right]$$

due to the projection's properties. Then

$$\mathbb{E}_k\left[\left\|x^{k+1} - x_*^{k+1}\right\|^2\right]$$

$$\leq \mathbb{E}_k\left[\left\|x^k - \gamma\frac{1}{n}\sum_{i=1}^n w_i^k \sum_{j=1}^{B_i^k} \nabla f_i(x^k;\xi_{ij}^k) - x_*^k\right\|^2\right]$$

$$= \left\|x^k - x_*^k\right\|^2 - 2\gamma\mathbb{E}_k\left[\left\langle x^k - x_*^k, \frac{1}{n}\sum_{i=1}^n w_i^k \sum_{j=1}^{B_i^k} \nabla f_i(x^k;\xi_{ij}^k)\right\rangle\right] + \gamma^2\mathbb{E}_k\left[\left\|\frac{1}{n}\sum_{i=1}^n w_i^k \sum_{j=1}^{B_i^k} \nabla f_i(x^k;\xi_{ij}^k)\right\|^2\right].$$

Using unbiasedness (Assumption 1.2) and the variance decomposition equality, we get

$$\mathbb{E}_k\left[\left\|x^{k+1} - x_*^{k+1}\right\|^2\right]$$

$$\leq \left\|x^k - x_*^k\right\|^2 - 2\gamma\left\langle x^k - x_*^k, \frac{1}{n}\sum_{i=1}^n w_i^k B_i^k \nabla f_i(x^k)\right\rangle$$

$$+ \gamma^2\left\|\frac{1}{n}\sum_{i=1}^n w_i^k B_i^k \nabla f_i(x^k)\right\|^2 + \gamma^2\mathbb{E}_k\left[\left\|\frac{1}{n}\sum_{i=1}^n w_i^k \sum_{j=1}^{B_i^k}\left(\nabla f_i(x^k;\xi_{ij}^k) - \nabla f_i(x^k)\right)\right\|^2\right].$$

Consider the last term, due to the independence of stochastic gradients and Assumption 1.2, we ensure that

$$\mathbb{E}_k\left[\left\|\frac{1}{n}\sum_{i=1}^n w_i^k \sum_{j=1}^{B_i^k}\left(\nabla f_i(x^k;\xi_{ij}^k) - \nabla f_i(x^k)\right)\right\|^2\right]$$

---

[3]It is well-know that Assumption 5.3 implies Assumption 1.1. In Section F, we prove that Assumptions 5.1, 5.6 and Assumption 5.7 with constant $\mu$ imply Assumption 5.2 with constant $\mu/4$.

$$= \frac{1}{n^2} \sum_{i=1}^{n} (w_i^k)^2 \sum_{j=1}^{B_i^k} \mathbb{E}_k \left[ \left\| \nabla f_i(x^k; \xi_{ij}^k) - \nabla f_i(x^k) \right\|^2 \right] \leq \frac{1}{n^2} \sum_{i=1}^{n} (w_i^k)^2 B_i^k \sigma^2.$$

Thus

$$\mathbb{E}_k \left[ \left\| x^{k+1} - x_*^{k+1} \right\|^2 \right] \tag{11}$$

$$\leq \left\| x^k - x_*^k \right\|^2 - \frac{2\gamma}{n} \sum_{i=1}^{n} w_i^k B_i^k \left\langle x^k - x_*^k, \nabla f_i(x^k) \right\rangle + \gamma^2 \left\| \frac{1}{n} \sum_{i=1}^{n} w_i^k B_i^k \nabla f_i(x^k) \right\|^2 + \frac{\gamma^2}{n^2} \sum_{i=1}^{n} (w_i^k)^2 B_i^k \sigma^2.$$

We now consider the second and the third term. Since $\frac{1}{n} \sum_{i=1}^{n} w_i^k B_i^k = 1$, using Jensen's inequality, we get

$$- 2\gamma \left\langle x^k - x_*^k, \frac{1}{n} \sum_{i=1}^{n} w_i^k B_i^k \nabla f_i(x^k) \right\rangle + \gamma^2 \left\| \frac{1}{n} \sum_{i=1}^{n} w_i^k B_i^k \nabla f_i(x^k) \right\|^2$$

$$\leq -2\gamma \left\langle x^k - x_*^k, \frac{1}{n} \sum_{i=1}^{n} w_i^k B_i^k \nabla f_i(x^k) \right\rangle + \gamma^2 \frac{1}{n} \sum_{i=1}^{n} w_i^k B_i^k \left\| \nabla f_i(x^k) \right\|^2.$$

Due to Assumption 5.6, we get

$$\frac{1}{n} \sum_{i=1}^{n} w_i^k B_i^k \left\| \nabla f_i(x^k) \right\|^2 = \frac{1}{n} \sum_{i=1}^{n} w_i^k B_i^k \left\| \nabla f_i(x^k) - \nabla f_i(x_*^k) \right\|^2$$

$$\leq \frac{1}{n} \sum_{i=1}^{n} w_i^k B_i^k L_i \left\langle x^k - x_*^k, \nabla f_i(x^k) - \nabla f_i(x_*^k) \right\rangle$$

$$\leq L_{\max} \frac{1}{n} \sum_{i=1}^{n} w_i^k B_i^k \left\langle x^k - x_*^k, \nabla f_i(x^k) - \nabla f_i(x_*^k) \right\rangle$$

$$= L_{\max} \frac{1}{n} \sum_{i=1}^{n} w_i^k B_i^k \left\langle x^k - x_*^k, \nabla f_i(x^k) \right\rangle.$$

In the first inequality, we use Lemma C.1 under Assumption 5.3 and convexity (Assumption 5.1). In the second inequality, we use the bound $L_i \leq L_{\max}$ for all $i \in [n]$. Taking $\gamma \leq 1/L_{\max}$ and substituting the last inequality to (11), we obtain

$$\mathbb{E}_k \left[ \left\| x^{k+1} - x_*^{k+1} \right\|^2 \right]$$

$$\leq \left\| x^k - x_*^k \right\|^2 - (2\gamma - L_{\max}\gamma^2) \frac{1}{n} \sum_{i=1}^{n} w_i^k B_i^k \left\langle x^k - x_*^k, \nabla f_i(x^k) \right\rangle + \frac{\gamma^2}{n^2} \sum_{i=1}^{n} (w_i^k)^2 B_i^k \sigma^2$$

$$\leq \left\| x^k - x_*^k \right\|^2 - \gamma \frac{1}{n} \sum_{i=1}^{n} w_i^k B_i^k \left\langle x^k - x_*^k, \nabla f_i(x^k) \right\rangle + \frac{\gamma^2}{n^2} \sum_{i=1}^{n} (w_i^k)^2 B_i^k \sigma^2.$$

Using the convexity, Assumption 5.7, and Lemma C.2, we get

$$\mathbb{E}_k \left[ \left\| x^{k+1} - x_*^{k+1} \right\|^2 \right]$$

$$\leq \left\| x^k - x_*^k \right\|^2 - \frac{\gamma\mu}{2} \frac{1}{n} \sum_{i=1}^{n} w_i^k B_i^k \left\| x^k - x_*^k \right\|^2 + \frac{\gamma^2}{n^2} \sum_{i=1}^{n} (w_i^k)^2 B_i^k \sigma^2.$$

We take $w_i^k = n/\sum_{i=1}^{n} B_i^k$ in the theorem for all $i \in [n]$. Thus

$$\mathbb{E}_k \left[ \left\| x^{k+1} - x_*^{k+1} \right\|^2 \right] \leq \left( 1 - \frac{\gamma\mu}{2} \right) \left\| x^k - x_*^k \right\|^2 + \frac{\gamma^2 \sigma^2}{\sum_{i=1}^{n} B_i^k}.$$

In Algorithm 1, with the chosen weights $\{w_i^k\}$, we wait for the moment when $\sum_{i=1}^{n} B_i^k > S$. Thus

$$\mathbb{E}_k \left[ \left\| x^{k+1} - x_*^{k+1} \right\|^2 \right] \leq \left( 1 - \frac{\gamma\mu}{2} \right) \left\| x^k - x_*^k \right\|^2 + \frac{\gamma^2 \sigma^2}{S}.$$

Unrolling the recursion and taking the full expectation, we obtain

$$\mathbb{E}\left[\left\|x^{k+1} - x_*^{k+1}\right\|^2\right] \leq \left(1 - \frac{\gamma\mu}{2}\right)^{k+1}\left\|x^0 - x_*^0\right\|^2 + \sum_{j=0}^{k}\left(1 - \frac{\gamma\mu}{2}\right)^j \frac{\gamma^2\sigma^2}{S}$$

$$\leq \left(1 - \frac{\gamma\mu}{2}\right)^{k+1}\left\|x^0 - x_*^0\right\|^2 + \frac{2\gamma\sigma^2}{\mu S}.$$

Due the choice of $\gamma$, $S$, and the condition on $k$, we have $\mathbb{E}\left[\left\|x^{k+1} - x_*^{k+1}\right\|^2\right] \leq \varepsilon$.

It is sufficient to run the method for

$$\mathcal{O}\left(\frac{L_{\max}}{\mu}\log\frac{R^2}{\varepsilon}\right)$$

iterations. In each iteration, the method has to ensure that $\sum_{i=1}^n B_i^k > S$. A sufficient time for that is

$$2\min_{m\in[n]}\left[\left(\frac{1}{m}\sum_{i=1}^m \frac{1}{\tau_i}\right)^{-1}\left(1 + \frac{4\sigma^2}{mL_{\max}\varepsilon\mu}\right)\right].$$

under the fixed computation model (see Theorem 11 in (Tyurin et al., 2024)). □

**Theorem A.1.** *Consider the assumptions, algorithm, and parameters from Theorem 5.10. Then,* Rennala SGD *converges after at most* $\bar{t}_{\left\lceil c\times\frac{L_{\max}}{\mu}\log\frac{R^2}{\varepsilon}\right\rceil}$ *seconds, where the sequence* $\{\bar{t}_k\}$ *is defined recursively as* $\bar{t}_k :=$

$$\min\left\{t \geq 0 : \sum_{i=1}^n\left\lfloor\int_{\bar{t}_{k-1}}^t v_i(\tau)d\tau\right\rfloor \geq \max\left\{\left\lceil\frac{\sigma^2}{\varepsilon}\right\rceil, 1\right\}\right\} \tag{10}$$

*for all* $k \geq 1$ ($\bar{t}_0 \equiv 0$), *and* $c$ *is a universal constant.*

*Proof.* From the proof of Theorem 5.10, we know that it is sufficient to run the method for

$$c \times \frac{L_{\max}}{\mu}\log\frac{R^2}{\varepsilon}$$

iterations, where $c$ is a universal constant. The method waits the moment when $\sum_{i=1}^n B_i^k > S$ in each iteration. The workers work in parallel, and for all $i \in [n]$, will calculate

$$\left\lfloor\int_0^t v_i(\tau)d\tau\right\rfloor$$

stochastic gradients after $t$ seconds. In total, all workers will calculate $\sum_{i=1}^n\left\lfloor\int_0^t v_i(\tau)d\tau\right\rfloor$ stochastic gradients. Hence, the first iteration will end after

$$\bar{t}_1 := \min\left\{t \geq 0 : \sum_{i=1}^n\left\lfloor\int_0^t v_i(\tau)d\tau\right\rfloor \geq 2S\right\},$$

seconds. After that, the second iteration starts before time $\bar{t}_1$ and ends at least at time

$$\bar{t}_2 := \min\left\{t \geq 0 : \sum_{i=1}^n\left\lfloor\int_{\bar{t}_1}^t v_i(\tau)d\tau\right\rfloor \geq 2S\right\},$$

because worker $i$ can calculate at least

$$\left\lfloor\int_{\bar{t}_1}^t v_i(\tau)d\tau\right\rfloor$$

stochastic gradients between the end of the first iteration and a time $t$. Using the same reasoning, we can recursively define

$$\bar{t}_3, \ldots, \bar{t}_{\left\lceil c\times\frac{L_{\max}}{\mu}\log\frac{R^2}{\varepsilon}\right\rceil}.$$

The algorithm will converge after $\bar{t}_{\left\lceil c\times\frac{L_{\max}}{\mu}\log\frac{R^2}{\varepsilon}\right\rceil}$ seconds due to the discussion at the beginning of the theorem. □

**Theorem 5.8.** *Consider the* Weighted SGD *method. Let us fix any* $\varepsilon, L_{\max}, R, \mu, \sigma^2 > 0$ *such that* $\mu < L_{\max}/(2n)$, $\varepsilon < 0.01$, *and* $R > 10$. *For all* $B_1, \ldots, B_n \geq 0$ *and any possible choice of weights* $\{w_i(B_1, \ldots, B_n)\}$ *as functions of* $B_1, \ldots, B_n$, *there exist functions* $\{f_i\}$ *and stochastic gradients* $\{\nabla f_i(\cdot; \cdot)\}$ *such that* $\{f_i\}$ *satisfy Assumptions* 5.1, 5.3, 5.4, *and* 5.7 (*do not satisfy Assumption 5.6 in general*), *f satisfies Assumptions 5.2 and 1.1 with* $L = L_{\max}$, $\{\nabla f_i(\cdot; \cdot)\}$ *satisfy Assumption 1.2 such that the method requires at least*

$$\Omega\left(\left(\frac{1}{n}\sum_{i=1}^{n}\tau_i\right)\frac{\sigma^2}{\varepsilon n\mu^2}\log\left(\frac{R^2}{\varepsilon}\right)\right)$$

*seconds to find* $\varepsilon$*–solution in terms of distances to the solution set, when the method starts at a point in a distance less or equal to* $R$ *to the closest solution.*

*Proof.* If $w_i(B_1, \ldots, B_n) \times B_i = 1$ for all $i \in [n]$, then it is true since the method reduces to Malenia SGD. Assume that there exists a combination $B_1, \ldots, B_n$ such that $w_i(B_1, \ldots, B_n) \times B_i \neq 1$ for some $i \in [n]$. Let us define the shortcut $w_i(B_1, \ldots, B_n) \equiv w_i$ and fix this combination. Let us find the index of the weight with the smallest value, i.e., $j = \arg\min_{i \in [n]} w_i B_i$. Without loss of generality, assume that $j = 1$.

For all $i \in [n]$, we take the quadratic function $f_i : \mathbb{R}^2 \to \mathbb{R}$ such that $f_i(x, y) = 0.5 L_{\max} y^2$ for all $i \neq 1$ and $f_1(x, y) = 0.5\mu n x^2 + 0.5 L_{\max} y^2$, and the stochastic gradients $\nabla f_i(x, y) + [\xi_i, 0]^\top$ for all $i \in [n]$, where $\xi_1, \ldots, \xi_n$ are i.i.d. gaussian noises from $\mathcal{N}(0, \sigma^2)$.

One can easily check that these functions satisfy the assumptions from the theorem. Let us consider the second argument of the functions and consider Weighted SGD:

$$y^{k+1} = y^k - \frac{\gamma}{n}\sum_{i=1}^{n} w_i B_i L_{\max} y^k = (1 - \gamma L_{\max}) y^k,$$

where we simplify due to the agreement $\frac{1}{n}\sum_{i=1}^{n} w_i B_i = 1$. The sequence $y^k$ converges if $\gamma \leq 1/L_{\max}$. It is necessary to assume that $\gamma \leq 1/L_{\max}$. Let us now consider the sequence w.r.t. the first coordinate:

$$x^{k+1} = x^k - \frac{\gamma}{n}\left(w_1\left(B_1\mu n x^k + \sum_{j=1}^{B_1}\xi_{1,j}^k\right) + \sum_{i=2}^{n} w_i\left(\sum_{j=1}^{B_i}\xi_{i,j}^k\right)\right)$$

$$= (1 - \gamma w_1 B_1 \mu)x^k - \frac{\gamma}{n}\sum_{i=1}^{n} w_i\left(\sum_{j=1}^{B_i}\xi_{i,j}^k\right).$$

Notice that $\frac{1}{n}\sum_{i=1}^{n} w_i\left(\sum_{j=1}^{B_i}\xi_{i,j}^k\right) \sim \mathcal{N}\left(0, \frac{1}{n^2}\sum_{i=1}^{n} w_i^2 B_i \sigma^2\right)$. Therefore

$$\mathbb{E}_k\left[\left|x^{k+1}\right|^2\right] = (1 - \gamma w_1 B_1 \mu)^2\left|x^k\right|^2 + \frac{\gamma^2}{n^2}\sum_{i=1}^{n} w_i^2 B_i \sigma^2.$$

Necessarily, $w_1 B_1 > 0$. Otherwise, $\mathbb{E}\left[\left|x^{k+1}\right|^2\right] \geq \mathbb{E}\left[\left|x^k\right|^2\right] \geq \left|x^0\right|^2$ for all $k \geq 1$. Unrolling the recursion and taking the full expectation, we obtain

$$\mathbb{E}\left[\left|x^{k+1}\right|^2\right] = (1 - \gamma w_1 B_1 \mu)^{2k}\left|x^0\right|^2 + \sum_{j=0}^{k}(1 - \gamma w_1 B_1 \mu)^j\frac{\gamma^2}{n^2}\sum_{i=1}^{n} w_i^2 B_i \sigma^2$$

Since $\frac{1}{n}\sum_{i=1}^{n} w_i B_i = 1$ and $\gamma \leq 1/L_{\max} < 1/(2n\mu)$, we have $0 < \gamma w_1 B_1 \mu \leq 1/2$ and

$$\mathbb{E}\left[\left|x^{k+1}\right|^2\right] = (1 - \gamma w_1 B_1 \mu)^{2k}\left|x^0\right|^2 + \frac{1 - (1 - \gamma w_1 B_1 \mu)^{k+1}}{w_1 B_1}\frac{\gamma}{n^2\mu}\sum_{i=1}^{n} w_i^2 B_i \sigma^2. \quad (12)$$

In order to ensure that $\mathbb{E}\left[\left|x^{k+1}\right|^2\right] \leq \varepsilon$, Weighted SGD should do at least

$$k \geq \frac{1}{2\log(1 - \gamma w_1 B_1 \mu)}\log\left(\frac{\varepsilon}{|x^0|^2}\right) \geq \frac{1}{4\gamma w_1 B_1 \mu}\log\left(\frac{|x^0|^2}{\varepsilon}\right) \quad (13)$$

steps to get $(1 - \gamma w_1 B_1 \mu)^{2k} |x^0|^2 \leq \varepsilon$. Note that $k \geq 1$ because $|x^0|^2 \geq 1$ and $\varepsilon \leq 0.01$. Thus we can bound the second term in (12) in the following way:

$$\frac{1 - (1 - \gamma w_1 B_1 \mu)^{k+1}}{w_1 B_1} \frac{\gamma}{n^2 \mu} \sum_{i=1}^n w_i^2 B_i \sigma^2 \geq \frac{1}{2 w_1 B_1} \frac{\gamma}{n^2 \mu} \sum_{i=1}^n w_i^2 B_i \sigma^2$$

because $(1 - \gamma w_1 B_1 \mu)^k \leq \sqrt{\frac{\varepsilon}{|x^0|^2}} \leq \frac{1}{2}$. Therefore, it is necessary in the algorithm choose the parameters in a such way that $\frac{1}{2 w_1 B_1} \frac{\gamma}{n^2 \mu} \sum_{i=1}^n w_i^2 B_i \sigma^2 \leq \varepsilon$. Using this bound and (13), we obtain

$$k \geq \frac{\sigma^2}{8 B_1^2 \varepsilon n^2 \mu^2} \sum_{i=1}^n \frac{w_i^2}{w_1^2} B_i \log\left(\frac{|x^0|^2}{\varepsilon}\right) = \frac{\sigma^2}{8 \varepsilon n \mu^2} \frac{1}{n} \sum_{i=1}^n \frac{w_i^2 B_i^2}{w_1^2 B_1^2} \frac{1}{B_i} \log\left(\frac{|x^0|^2}{\varepsilon}\right).$$

Recall that $w_1^2 B_1^2 \leq w_i^2 B_i^2$ for all $i \in [n]$. Thus, the algorithm has to do

$$k \geq \frac{\sigma^2}{8 \varepsilon n \mu^2} \frac{1}{n} \sum_{i=1}^n \frac{1}{B_i} \log\left(\frac{|x^0|^2}{\varepsilon}\right).$$

steps. Taking $x^0 = R/\sqrt{2}$ and $y^0 = R/\sqrt{2}$, we get $\mathbb{E}\left[|x^k|^2 + |y^k|^2\right] \geq \varepsilon$ and

$$\mathbb{E}\left[f(x^k, y^k)\right] = \mathbb{E}\left[0.5 \mu \times (x^k)^2 + 0.5 L_{\max} \times (y^k)^2\right] \geq 0.5 \mu \varepsilon$$

for all

$$k < \frac{\sigma^2}{8 \varepsilon n \mu^2} \frac{1}{n} \sum_{i=1}^n \frac{1}{B_i} \log\left(\frac{|x^0|^2}{\varepsilon}\right).$$

Under the fixed computation model, $B_i = \left\lfloor \frac{t^*}{\tau_i} \right\rfloor$, where $t^*$ is the time of one iteration. We can conclude that the required total time is at least

$$t^* \times \frac{\sigma^2}{8 \varepsilon n \mu^2} \frac{1}{n} \sum_{i=1}^n \frac{1}{B_i} \log\left(\frac{|x^0|^2}{\varepsilon}\right) \geq \frac{\sigma^2}{8 \varepsilon n \mu^2} \frac{1}{n} \sum_{i=1}^n \tau_i \log\left(\frac{|x^0|^2}{\varepsilon}\right).$$

$\square$

**Theorem 5.5.** *Consider the* Weighted SGD *method. Let us fix any $\varepsilon, L_{\max}, R, \mu, \sigma^2 > 0$ such that $\mu < L_{\max}/(2n), \varepsilon < 0.01$, and $R > 10$. For all $B_1, \ldots, B_n \geq 0$ and any possible choice of weights $\{w_i(B_1, \ldots, B_n)\}$ as functions of $B_1, \ldots, B_n$, there exist functions $\{f_i\}$ and stochastic gradients $\{\nabla f_i(\cdot; \cdot)\}$ such that $\{f_i\}$ satisfy Assumptions 5.1, 5.3, and 5.4, $f$ satisfies Assumptions 5.2 and 1.1 with $L = L_{\max}$, $\{\nabla f_i(\cdot; \cdot)\}$ satisfy Assumption 1.2 such that the method requires at least*

$$\Omega\left(\left(\frac{1}{n} \sum_{i=1}^n \tau_i\right) \frac{\sigma^2}{\varepsilon n \mu^2} \log\left(\frac{R^2}{\varepsilon}\right)\right)$$

*seconds to find $\varepsilon$−solution in terms of distances to the solution set, when the method starts at a point in a distance less or equal to $R$ to the closest solution.*

*Proof.* The theorem is a simple corollary of Theorem 5.8 because Theorem 5.8 is stated under more strict assumptions on the class of the functions and stochastic gradients. The result of Theorem 5.8 holds even under additional Assumption 5.7. $\square$

**Theorem 5.9.** *Consider the* Weighted SGD *method. Let us fix any $\varepsilon, L_{\max}, R, \sigma^2 > 0$ such that $\mu < L_{\max}/(2n), \varepsilon < 0.01$, and $R > 10$. For all $B_1, \ldots, B_n \geq 1$ and any possible choice of weights $\{w_i(B_1, \ldots, B_n)\}$ as functions of $B_1, \ldots, B_n$, there exist functions $\{f_i\}$ and stochastic gradients $\{\nabla f_i(\cdot; \cdot)\}$ such that $\{f_i\}$ satisfy Assumptions 5.1, 5.3, 5.4, and 5.6 (**do not satisfy Assumption 5.7 in general**), $f$ satisfy Assumptions 5.2 and 1.1 with $L = L_{\max}$, $\{\nabla f_i(\cdot; \cdot)\}$ satisfy Assumption 1.2, such that the method requires at least*

$$\Omega\left(\left(\frac{1}{n} \sum_{i=1}^n \tau_i\right) \frac{\sigma^2}{\varepsilon n \mu^2} \log\left(\frac{R^2}{\varepsilon}\right)\right)$$

*seconds to find $\varepsilon$−solution in terms of distances to the solution set, when the method starts at a point in a distance less or equal to $R$ to the closest solution.*

*Proof.* Let us define the shortcut $w_i(B_1, \ldots, B_n) \equiv w_i$. In this construction, we consider the function $f_i : \mathbb{R}^2 \to \mathbb{R}$ such that

$$f_i(x, y) = \frac{\frac{\mu}{B_i}}{\frac{2}{n} \sum_{i=1}^{n} \frac{1}{B_i}} x^2 + \frac{L_{\max}}{2} y^2.$$

and the stochastic gradients $\nabla f_i(x, y) + [\xi_i, 0]^\top$ for all $i \in [n]$, where $\xi_1, \ldots, \xi_n$ are i.i.d. gaussian noises from $\mathcal{N}(0, \sigma^2)$. One can easily check that the assumptions from the theorem hold. Using the same reasoning as in the proof of Theorem 5.8, we have to take $\gamma \leq 1/L_{\max}$. Next, we consider the sequence of Weighted SGD w.r.t. the first argument:

$$x^{k+1} = x^k - \gamma \left( \frac{\sum_{i=1}^{n} w_i \mu}{\sum_{i=1}^{n} \frac{1}{B_i}} x^k + \frac{1}{n} \sum_{i=1}^{n} w_i \left( \sum_{j=1}^{B_i} \xi_{i,j}^k \right) \right)$$

$$= \left( 1 - \gamma \mu \frac{\sum_{i=1}^{n} w_i}{\sum_{i=1}^{n} \frac{1}{B_i}} \right) x^k - \frac{\gamma}{n} \sum_{i=1}^{n} w_i \left( \sum_{j=1}^{B_i} \xi_{i,j}^k \right).$$

Note that $\frac{\sum_{i=1}^{n} w_i}{\sum_{i=1}^{n} \frac{1}{B_i}} \leq \sum_{i=1}^{n} w_i B_i = n$ due to $\frac{1}{n} \sum_{i=1}^{n} w_i B_i = 1$, and $\gamma \leq 1/L_{\max} \leq 1/(2n\mu)$. Using the same reasoning as in the proof of Theorem 5.8, we get

$$\mathbb{E}\left[ \left| x^{k+1} \right|^2 \right] = \left( 1 - \gamma \mu \frac{\sum_{i=1}^{n} w_i}{\sum_{i=1}^{n} \frac{1}{B_i}} \right)^{2k} \left| x^0 \right|^2 + \frac{1 - \left( 1 - \gamma \mu \frac{\sum_{i=1}^{n} w_i}{\sum_{i=1}^{n} \frac{1}{B_i}} \right)^{k+1}}{\frac{\sum_{i=1}^{n} w_i}{\sum_{i=1}^{n} \frac{1}{B_i}}} \frac{\gamma}{n^2 \mu} \sum_{i=1}^{n} w_i^2 B_i \sigma^2.$$

$$(14)$$

Weighted SGD should do at least

$$k \geq \frac{1}{4\gamma \mu \frac{\sum_{i=1}^{n} w_i}{\sum_{i=1}^{n} \frac{1}{B_i}}} \log \left( \frac{\left| x^0 \right|^2}{\varepsilon} \right) \tag{15}$$

steps to get $\left( 1 - \gamma \mu \frac{\sum_{i=1}^{n} w_i}{\sum_{i=1}^{n} \frac{1}{B_i}} \right)^{2k} \left| x^0 \right|^2 \leq \varepsilon$. The parameters should satisfy the inequality

$$\frac{1}{\frac{\sum_{i=1}^{n} w_i}{\sum_{i=1}^{n} \frac{1}{B_i}}} \frac{\gamma}{2n^2 \mu} \sum_{i=1}^{n} w_i^2 B_i \sigma^2 \leq \varepsilon$$

to ensure that $\mathbb{E}\left[ \left| x^{k+1} \right|^2 \right] \leq \varepsilon$ for some $k \geq 0$. Combining this inequality with (15), we get

$$k \geq \left( \sum_{i=1}^{n} \frac{1}{B_i} \right)^2 \frac{\sigma^2}{8n^2 \mu^2 \varepsilon} \frac{\sum_{i=1}^{n} w_i^2 B_i}{\left( \sum_{i=1}^{n} w_i \right)^2} \log \left( \frac{\left| x^0 \right|^2}{\varepsilon} \right).$$

It is left to use the Cauchy–Schwarz inequality $\frac{\sum_{i=1}^{n} w_i^2 B_i}{\left( \sum_{i=1}^{n} w_i \right)^2} \geq \left( \sum_{i=1}^{n} \frac{1}{B_i} \right)^{-1}$ to ensure that the method will not find an $\varepsilon$–solution in terms of the distance to the solution $x_* = 0$ before

$$\frac{1}{8} \frac{\sigma^2}{\varepsilon n \mu^2} \left( \frac{n}{\sum_{i=1}^{n} \frac{1}{B_i}} \right)^{-1} \log \left( \frac{\left| x^0 \right|^2}{\varepsilon} \right)$$

steps. Since $f(x) = \frac{\mu}{2} x^2 + \frac{L_{\max}}{2} y^2$, the method will need at least

$$\frac{1}{16} \frac{\sigma^2}{\varepsilon n \mu} \left( \frac{n}{\sum_{i=1}^{n} \frac{1}{B_i}} \right)^{-1} \log \left( \frac{\mu \left| x^0 \right|^2}{\varepsilon} \right)$$

---

**Protocol 2** Time Multiple Oracles Protocol

---

1: **Input:** function(s) $f \in \mathcal{F}$, oracles and distributions $((O_1, ..., O_n), (\mathcal{D}_1, ..., \mathcal{D}_n)) \in \mathcal{O}(f)$,
   algorithm $A \in \mathcal{A}$
2: $s_i^0 = 0$ for all $i \in [n]$
3: **for** $k = 0, \ldots, \infty$ **do**
4: $\quad (t^{k+1}, i^{k+1}, x^k) = A^k(g^1, \ldots, g^k),$ $\qquad\qquad\qquad\qquad\quad \triangleright t^{k+1} \geq t^k$
5: $\quad (s_{i^{k+1}}^{k+1}, g^{k+1}) = O_{i^{k+1}}(t^{k+1}, x^k, s_{i^{k+1}}^k, \xi^{k+1}), \quad \xi^{k+1} \sim \mathcal{D}_{i^{k+1}} \quad \triangleright s_j^{k+1} = s_j^k \quad \forall j \neq i^{k+1}$
6: **end for**

---

iterations to find an $\varepsilon$–solution in terms of function values. As in the proof of Theorem 5.8, it is sufficient to take $x^0 = R/\sqrt{2}$ and $y^0 = R/\sqrt{2}$. Under the fixed computation model, $B_i = \left\lfloor \frac{t^*}{\tau_i} \right\rfloor$, where $t^*$ is the time of one iteration. We can conclude that the required total time is at least

$$t^* \times \frac{1}{16} \frac{\sigma^2}{\varepsilon n \mu} \left( \frac{n}{\sum_{i=1}^n \frac{1}{B_i}} \right)^{-1} \log \left( \frac{\mu \left| x^0 \right|^2}{\varepsilon} \right) \geq \frac{1}{16} \frac{\sigma^2}{\varepsilon n \mu} \frac{1}{n} \sum_{i=1}^n \tau_i \log \left( \frac{\mu \left| x^0 \right|^2}{\varepsilon} \right).$$

$\qquad\qquad\qquad\qquad\qquad\qquad\qquad\qquad\qquad\qquad\qquad\qquad\qquad\qquad\qquad\qquad\qquad\qquad\qquad$ □

## C  AUXILIARY RESULTS

In this section, we present well-known results from optimization.

**Lemma C.1** (Nesterov (2018)). *Let $f : \mathbb{R}^d \to \mathbb{R}$ be a function, which $L$–smooth and convex. Then for all $x, y \in \mathbb{R}^d$ we have:*

$$\|\nabla f(x) - \nabla f(y)\|^2 \leq L \langle \nabla f(x) - \nabla f(y), x - y \rangle. \tag{16}$$

**Lemma C.2** (Karimi et al. (2016)). *Let $f : \mathbb{R}^d \to \mathbb{R}$ be a convex function, which satisfies PŁ condition with a parameter $\mu$ (Assumption 5.2). Then, for all $x \in \mathbb{R}^d$, we have*

$$\langle \nabla f(x), x - \bar{x}_* \rangle \geq \frac{\mu}{2} \|\bar{x}_* - x\|^2 \tag{17}$$

*where $\bar{x}_*$ is the projection of $x$ onto the solution set of $\min_{x \in \mathbb{R}^d} f(x)$.*

## D  LOWER BOUND IN THE HETEROGENEOUS CONVEX SETTING

This section complements the results from (Tyurin & Richtárik, 2023), where the authors only prove the optimal time complexities in the *homogeneous nonconvex*, *heterogeneous nonconvex*, and *homogeneous convex* settings. Here, we resolve the last piece, the *heterogeneous convex* setting. Following Tyurin & Richtárik (2023), we have to formalize and introduce the following protocol and classes.

We investigate the optimization problem (1) when the function $f$ is convex. For the convex case, using Protocol 2, we use the complexity measure

$$\mathfrak{m}_{\text{time}}(\mathcal{A}, \mathcal{F}) := \inf_{A \in \mathcal{A}} \sup_{f \in \mathcal{F}} \sup_{(O, \mathcal{D}) \in \mathcal{O}(f)} \inf \left\{ t \geq 0 \, \middle| \, \mathbb{E} \left[ \inf_{k \in S_t} f(x^k) \right] - \inf_{x \in Q} f(x) \leq \varepsilon \right\},$$

$$S_t := \left\{ k \in \mathbb{N}_0 \middle| t^k \leq t \right\}, \tag{18}$$

where the sequences $t^k$ and $x^k$ are generated by Protocol 2, and $Q$ is a convex set. Let us take any set $Q$, and consider the following class of convex functions.

**Definition D.1** (Function Class $\mathcal{F}_{Q,L,M}^{\text{conv}}$).

We assume that a function $f : \mathbb{R}^d \to \mathbb{R}$ is convex, differentiable, $L$-smooth on the set $Q$, i.e.,

$$\|\nabla f(x) - \nabla f(y)\| \leq L \|x - y\| \quad \forall x, y \in Q,$$

and $M$-Lipschitz on the set $Q$, i.e.,

$$|f(x) - f(y)| \leq M \|x - y\| \quad \forall x, y \in Q.$$

A set of all functions with such properties we define as $\mathcal{F}_{Q,L,M}^{\text{conv}}$.

**Definition D.2** (Algorithm Class $\mathcal{A}_{\mathrm{zr}}$)**.**
An algorithm $A = \{A^k\}_{k=0}^{\infty}$ is a sequence such that

$$A^k \;:\; \underbrace{\mathbb{R}^d \times \cdots \times \mathbb{R}^d}_{k \text{ times}} \to \mathbb{R}_{\geq 0} \times \mathbb{R}^d \quad \forall k \geq 1, A^0 \in \mathbb{R}_{\geq 0} \times \mathbb{R}^d,$$

and, for all $k \geq 1$ and $g^1, \ldots, g^k \in \mathbb{R}^d$, $t^{k+1} \geq t^k$, where $t^{k+1}$ and $t^k$ are defined as $(t^{k+1}, \cdot) = A^k(g^1, \ldots, g^k)$ and $(t^k, \cdot) = A^{k-1}(g^1, \ldots, g^{k-1})$.

The following oracle helps to formalize the fixed computation model.

$$O_\tau^{\nabla f} \;:\; \underbrace{\mathbb{R}_{\geq 0}}_{\text{time}} \times \underbrace{\mathbb{R}^d}_{\text{point}} \times \underbrace{(\mathbb{R}_{\geq 0} \times \mathbb{R}^d \times \{0,1\})}_{\text{input state}} \times \mathbb{S}_\xi \to \underbrace{(\mathbb{R}_{\geq 0} \times \mathbb{R}^d \times \{0,1\})}_{\text{output state}} \times \mathbb{R}^d$$

$$\text{such that } O_\tau^{\nabla f}(t, x, (s_t, s_x, s_q), \xi) = \begin{cases} ((t, x, 1), & 0), & s_q = 0, \\ ((s_t, s_x, 1), & 0), & s_q = 1 \text{ and } t < s_t + \tau, \\ ((0, 0, 0), & \nabla f(s_x; \xi)), & s_q = 1 \text{ and } t \geq s_t + \tau, \end{cases} \tag{19}$$

and $\nabla f(\cdot; \cdot)$ is a stochastic mapping.

**Definition D.3** (Oracle Class $\mathcal{O}_{\tau_1,\ldots,\tau_n}^{\mathrm{conv}, \sigma^2}$)**.**
Let us consider an oracle class such that, for any $f \in \mathcal{F}_{Q,L,M}^{\mathrm{conv}}$, it returns oracles $O_i = O_{\tau_i}^{\nabla f_i}$ and distributions $\mathcal{D}_i$ for all $i \in [n]$, where $\nabla f_i(\cdot; \cdot)$ is an unbiased $\sigma^2$-variance-bounded mapping on the set $Q$ of the gradient of the local function in worker $i$. The oracles $O_{\tau_i}^{\nabla f_i}$ are defined in (19). We define such oracle class as $\mathcal{O}_{\tau_1,\ldots,\tau_n}^{\mathrm{conv}, \sigma^2}$. Without loss of generality, we assume that $0 < \tau_1 \leq \cdots \leq \tau_n$.

Notice that this oracle class differs from the oracle class for convex functions in (Tyurin & Richtárik, 2023) because we consider the heterogeneous setting where the oracles return unbiased stochastic gradients of the local functions $f_i$, which can be different. We refer the reader to (Tyurin & Richtárik, 2023) for additional details about the time complexities formalization. We are now ready to state the theorem.

**Theorem D.4** (Informal theorem (see the formal Theorem D.5))**.** *Let Assumptions 5.1, 1.1, and 1.2 hold. It is impossible to converge faster than*

$$\Theta \left( \tau_n \sqrt{L} R / \sqrt{\varepsilon} + \left( 1/n \sum_{i=1}^{n} \tau_i \right) \sigma^2 R^2 / n\varepsilon^2 \right)$$

*seconds under the fixed computation model.*

**Theorem D.5.** *Let us consider the oracle class $\mathcal{O}_{\tau_1,\ldots,\tau_n}^{\mathrm{conv}, \sigma^2}$ for some $\sigma^2 > 0$ and $0 < \tau_1 \leq \cdots \leq \tau_n$. We fix any $R, L, M, \varepsilon > 0$ such that $\sqrt{L}R > c_1 \sqrt{\varepsilon} > 0$ and $\sigma^2 \geq M^2$. In the view Protocol 2, for any algorithm $A \in \mathcal{A}_{\mathrm{zr}}$, there exists a set $Q$, a function $f \in \mathcal{F}_{Q,L,M}^{\mathrm{conv}}$ and oracles and distributions $((O_1, \ldots, O_n), (\mathcal{D}_1, \ldots, \mathcal{D}_n)) \in \mathcal{O}_{\tau_1,\ldots,\tau_n}^{\mathrm{conv}, \sigma^2}(f)$ such that*

$$\mathbb{E}\left[ \inf_{k \in S_t} f(x^k) \right] - \inf_{x \in Q} f(x) > \varepsilon,$$

*where $S_t := \left\{ k \in \mathbb{N}_0 \big| t^k \leq t \right\}$,*

$$t = c \times \left[ \tau_n \min \left\{ \frac{\sqrt{L}R}{\sqrt{\varepsilon}}, \frac{M^2 R^2}{\varepsilon^2} \right\} + \left( \frac{1}{n} \sum_{i=1}^{n} \tau_i \right) \frac{\sigma^2 R^2}{n\varepsilon^2} \right],$$

*and $R$ is the euclidean distance between $0$ (starting point) and the closest solution $x_* \in Q$. The quantities $c_1$, and $c$ are universal constants.*

*Proof. First term.* It is easy to prove the dependence on the first term $\tau_n \min\{\frac{\sqrt{L}R}{\sqrt{\varepsilon}}, \frac{M^2 R^2}{\varepsilon^2}\}$ using the same idea as in (Lu & De Sa, 2021; Tyurin & Richtárik, 2023; Huang et al., 2022). It is sufficient

to put a "hard" convex function (Nesterov, 2018; Woodworth et al., 2018) to the slowest worker corresponding with the time $\tau_n = \max_{i \in [n]} \tau_i$. In particular, we can consider the "hard" quadratic function $\bar{f}$ from (Nesterov, 2018)[Section 2.1.2] and take the functions

$$f_i(x) = \begin{cases} n \times \bar{f}(x), & i = n \\ 0, & i < n. \end{cases}$$

for all $x \in \mathbb{R}^d$. The function $f = \frac{1}{n}\sum_{i=1}^n f_i = \bar{f}$ belongs to the class $\mathcal{F}_{Q,L,\infty}^{\text{conv}}$. We take the stochastic gradients without noise, i.e., $\nabla f_i(x;\xi_i) = \nabla f_i(x)$ deterministically for all $x \in \mathbb{R}^d$, $\xi_i \in \mathbb{S}_{\xi_i}$, and $i \in [n]$. It is clear that the only worker that can solve the problem is worker $n$, and it takes $\tau_n$ seconds to find one gradient by the oracle construction. Thus, the required time complexity is $\Theta\left(\tau_n \frac{\sqrt{L}R}{\sqrt{\varepsilon}}\right)$ since the required oracle complexity is $\Theta\left(\frac{\sqrt{L}R}{\sqrt{\varepsilon}}\right)$ (Nesterov, 2018). One can get $\Theta\left(\tau_n \frac{M^2 R^2}{\varepsilon^2}\right)$ using the same reasoning.

*Second term.* The proof of the second term is slightly trickier and uses the construction from (Woodworth et al., 2018). Let us fix any algorithm. We use the proof of Lemma 10 from (Woodworth et al., 2018) that has the following result. For any $\sigma^2$, $B > 0$ and any algorithm, it is possible to construct a *one dimensional linear* function $g : \mathbb{R} \to \mathbb{R}$ on the domain $\{x \in \mathbb{R} : |x| \leq B\}$, a stochastic gradient mapping $\nabla g : \mathbb{R} \times \mathbb{S}_\xi \to \mathbb{R}$, and a distribution $\mathcal{D}$ such that

$$\mathbb{E}\left[g(x^N)\right] - \min_{|x| \leq B} g(x) \geq \frac{\sigma B}{8\sqrt{N}} \tag{20}$$

after $N$ queries of the oracle, where $\nabla g$ is unbiased and $\sigma^2$-variance-bounded.

The idea is to put $g$ to each worker but with different domain sizes. In particular, for all $i \in [n]$, we take the function $f_i : \mathbb{R}^n \to \mathbb{R}$ such that

$$f_i(x) = g(x_i), \tag{21}$$

where $g$ is the function from Lemma 10 of (Woodworth et al., 2018), and $x_i$ is the $i^{\text{th}}$ coordinate of a vector $x$. For all $i \in [n]$, we consider the function $f_i$ on the domain $\{x_i \in \mathbb{R} \mid |x_i| \leq R_i\}$, where $R_i := R \times \frac{\sqrt{\tau_i}}{\sqrt{\sum_{i=1}^n \tau_i}}$. One can see that $f$ is convex, $0$–smooth (because $g$ is linear), and $M$–Lipschitz (because $\sigma \leq M$). The distance between $0$ and the optimal point is less or equal to $R$ because

$$\sum_{i=1}^n R_i^2 = \sum_{i=1}^n R^2 \frac{\tau_i}{\sum_{i=1}^n \tau_i} = R^2$$

and the optimal point for the problem $g(x_i) \to \min_{|x_i| \leq R_i}$ is either $R_i$ or $-R_i$. We take

$$Q = \{x \in \mathbb{R}^n : |x_i| \leq R_i \quad \forall i \in [n]\}.$$

Let us define the time

$$\bar{t} := \frac{\sigma^2 R^2}{256 n \varepsilon^2}\left(\frac{1}{n}\sum_{i=1}^n \tau_i\right). \tag{22}$$

By the time $\bar{t}$, worker $i$ can calculate at most

$$N_i := \left\lfloor \frac{\bar{t}}{\tau_i} \right\rfloor \tag{23}$$

stochastic gradients. Therefore,

$$\mathbb{E}\left[f(\bar{x})\right] - \min_{x \in Q} f(x) = \frac{1}{n}\sum_{i=1}^n\left(\mathbb{E}\left[g(\bar{x}_i)\right] - \min_{|x_i| \leq B_i} g(\bar{x}_i)\right) \overset{(20)}{\geq} \frac{1}{n}\sum_{i=1}^n \frac{\sigma R_i}{8\sqrt{N_i}}$$

$$= \frac{1}{n}\sum_{i=1}^n \frac{\sigma R \sqrt{\tau_i}}{8\sqrt{N_i}\sqrt{\sum_{i=1}^n \tau_i}} \overset{(22),(23)}{\geq} \sum_{i=1}^n \frac{2\varepsilon\tau_i}{\sum_{i=1}^n \tau_i} = 2\varepsilon.$$

where $\bar{x} \in \mathbb{R}^n$ is any possible output of the algorithm before the time $\bar{t}$.

$\square$

# E   PROOF OF THEOREMS E.1 AND E.2

**Theorem E.1.** *Let Assumptions 5.1, 5.2, 1.2, 1.1 hold, **and the functions** $\{f_i\}$ **are equal**. Let us take* $\gamma = 1/L$ *and* $S = 4\sigma^2/\mu L \varepsilon$*, then* Rennala SGD *(Algorithm 1 with* $w_i^k = n/\sum_{i=1}^n B_i^k$*) finds* $x^{k+1}$ *such that* $\mathbb{E}\left[\left\|x^{k+1} - x_*^{k+1}\right\|^2\right] \leq \varepsilon$ *after*

$$\mathcal{O}\left(\min_{m \in [n]}\left[\left(\frac{1}{m}\sum_{i=1}^m \frac{1}{\tau_i}\right)^{-1}\left(\frac{L}{\mu} + \frac{\sigma^2}{m\varepsilon\mu^2}\right)\right]\log\frac{R^2}{\varepsilon}\right) \tag{24}$$

*seconds, where* $x_*^{k+1}$ *is the closest solution to* $x^{k+1}$*.*

*Proof.* Since the functions are equal, Rennala SGD is equivalent to

$$x^{k+1} = x^k - \gamma g_{\mathsf{R}}^k,$$

$$g_{\mathsf{R}}^k := \frac{1}{\sum_{i=1}^n B_i^k}\sum_{i=1}^n\sum_{j=1}^{B_i^k}\nabla f(x^k; \xi_{ij}^k).$$

Clearly, $g_{\mathsf{R}}^k$ is unbiased and

$$\mathbb{E}_k\left[\left\|g_{\mathsf{R}}^k - \nabla f(x^k)\right\|^2\right] = \left(\sum_{i=1}^n B_i^k\right)^{-2}\sum_{i=1}^n\sum_{j=1}^{B_i^k}\mathbb{E}_k\left[\left\|\nabla f(x^k; \xi_{ij}^k) - \nabla f(x^k)\right\|\right]^2 \leq \sigma^2\left(\sum_{i=1}^n B_i^k\right)^{-1}.$$

Rennala SGD waits for the moment when $\sum_{i=1}^n B_i^k > S$ (see Alg. 1 with $w_i^k = n/\sum_{i=1}^n B_i^k$). Thus

$$\mathbb{E}_k\left[\left\|g_{\mathsf{R}}^k - \nabla f(x^k)\right\|^2\right] \leq \frac{\sigma^2}{S} \leq \frac{\mu L \varepsilon}{4}$$

We can use Theorem E.3 to get

$$\mathbb{E}\left[\left\|x^{k+1} - x_*^{k+1}\right\|^2\right] \leq \left(1 - \frac{\gamma\mu}{2}\right)^{k+1}\left\|x^0 - x_*^k\right\|^2 + \frac{\gamma L \varepsilon}{2}.$$

Since $\gamma = \frac{1}{L}$, we obtain

$$\mathbb{E}\left[\left\|x^{k+1} - x_*^{k+1}\right\|^2\right] \leq \left(1 - \frac{\mu}{2L}\right)^{k+1}\left\|x^0 - x_*^k\right\|^2 + \frac{\varepsilon}{2}.$$

The last inequality ensure that the method finds an $\varepsilon$–solution after

$$\mathcal{O}\left(\frac{L}{\mu}\right)$$

iterations. In each iteration, the method has to ensure that $\sum_{i=1}^n B_i^k > S$. A sufficient time for that is

$$2\min_{m \in [n]}\left[\left(\frac{1}{m}\sum_{i=1}^m \frac{1}{\tau_i}\right)^{-1}(1 + S)\right].$$

under the fixed computation model (see Theorem 11 in (Tyurin et al., 2024)). It is left to multiply this time by $\mathcal{O}\left(\frac{L}{\mu}\right)$. □

**Theorem E.2.** *Let Assumptions 5.1, 5.2, 1.2, 1.1 hold. Let us take* $\gamma = 1/L$ *and* $S = 4\sigma^2/\mu L \varepsilon$*, then* Malenia SGD *(Algorithm 1 with* $w_i^k = 1/B_i^k$*) finds* $x^{k+1}$ *such that* $\mathbb{E}\left[\left\|x^{k+1} - x_*^{k+1}\right\|^2\right] \leq \varepsilon$ *after*

$$\mathcal{O}\left(\left[\tau_n\frac{L}{\mu} + \left(\frac{1}{n}\sum_{i=1}^n \tau_i\right)\frac{\sigma^2}{n\varepsilon\mu^2}\right]\log\frac{R^2}{\varepsilon}\right) \tag{25}$$

*seconds, where* $x_*^{k+1}$ *is the closest solution to* $x^{k+1}$*.*

*Proof.* The proof of this theorem almost repeats the proof of Theorem E.1. The variance of Malenia SGD is

$$\mathbb{E}_k\left[\left\|g_{\mathsf{M}}^k - \nabla f(x^k)\right\|^2\right] = \mathbb{E}_k\left[\left\|\frac{1}{n}\sum_{i=1}^n \frac{1}{B_i^k}\sum_{j=1}^{B_i^k}\nabla f_i(x^k;\xi_{ij}^k) - \nabla f(x^k)\right\|^2\right] \le \frac{\sigma^2}{n}\left(\frac{1}{n}\sum_{j=1}^n \frac{1}{B_i^k}\right).$$

The method waits for the moment when $\left(\frac{1}{n}\sum_{i=1}^n \frac{1}{B_i^k}\right)^{-1} > \frac{S}{n}$. Therefore

$$\mathbb{E}_k\left[\left\|g_{\mathsf{M}}^k - \nabla f(x^k)\right\|^2\right] \le \frac{\sigma^2}{S}.$$

Using the same reasoning, the method finds an $\varepsilon$−solution after

$$\mathcal{O}\left(\frac{L}{\mu}\right)$$

iterations. In each iteration, the method has to ensure that $\left(\frac{1}{n}\sum_{i=1}^n \frac{1}{B_i^k}\right)^{-1} > \frac{S}{n}$. A sufficient time for that is

$$\bar{t} = 2\left(\tau_n + \left(\frac{1}{n}\sum_{i=1}^n \tau_i\right)\frac{S}{n}\right)$$

under the fixed computation model because the number of computed stochastic gradients $B_i^k \ge \left\lfloor\frac{\bar{t}}{\tau_i}\right\rfloor$, and

$$\frac{1}{n}\sum_{i=1}^n \frac{1}{B_i^k} \le \frac{1}{n}\sum_{i=1}^n \frac{1}{\left\lfloor\frac{\bar{t}}{\tau_i}\right\rfloor} \le \frac{1}{n}\sum_{i=1}^n \frac{2\tau_i}{\bar{t}} < \frac{n}{S},$$

where we use $\lfloor x \rfloor \ge \frac{x}{2}$ for all $x \ge 1$. Multiplying $\bar{t}$ by $\mathcal{O}\left(\frac{L}{\mu}\right)$, we get the result. $\square$

**Theorem E.3.** *Consider the method*

$$x^{k+1} = x^k - \gamma\nabla f(x^k;\xi^k), \tag{26}$$

*where* $\mathbb{E}_k\left[\nabla f(x^k;\xi^k)\right] = \nabla f(x^k)$, $\mathbb{E}_k\left[\left\|\nabla f(x^k;\xi^k) - \nabla f(x^k)\right\|^2\right] \le \sigma^2$, *and* $\sigma^2 > 0$. *Let Assumptions 5.1, 5.2, and 1.1 hold. Let us take* $\gamma = 1/L$ *and* $S = 2\sigma^2/\mu L\varepsilon$, *then the method finds* $x^{k+1}$ *such that*

$$\mathbb{E}\left[\left\|x^{k+1} - x_*^{k+1}\right\|^2\right] \le \left(1 - \frac{\gamma\mu}{2}\right)^{k+1}\left\|x^0 - x_*^k\right\|^2 + \frac{2\gamma\sigma^2}{\mu},$$

*where* $x_*^{k+1}$ *is the closest solution of* $\min_{x\in\mathbb{R}^d} f(x)$ *to* $x^{k+1}$.

*Proof.* Using the properties of the projection and (26), we have

$$\mathbb{E}_k\left[\left\|x^{k+1} - x_*^{k+1}\right\|^2\right] \le \mathbb{E}_k\left[\left\|x^{k+1} - x_*^k\right\|^2\right]$$

$$= \mathbb{E}_k\left[\left\|x^k - \gamma\nabla f(x^k;\xi^k) - x_*^k\right\|^2\right]$$

$$= \mathbb{E}_k\left[\left\|x^k - x_*^k\right\|^2\right] - 2\gamma\mathbb{E}_k\left[\left\langle\nabla f(x^k;\xi^k), x^k - x_*^k\right\rangle\right] + \gamma^2\mathbb{E}_k\left[\left\|\nabla f(x^k;\xi^k)\right\|^2\right]$$

$$= \mathbb{E}_k\left[\left\|x^k - x_*^k\right\|^2\right] - 2\gamma\left\langle\nabla f(x^k), x^k - x_*^k\right\rangle + \gamma^2\mathbb{E}_k\left[\left\|\nabla f(x^k;\xi^k)\right\|^2\right].$$

In the last equality, we use the unbiasedness. Due the variance decomposition equality, we get

$$\mathbb{E}_k\left[\left\|x^{k+1} - x_*^{k+1}\right\|^2\right] \le \mathbb{E}_k\left[\left\|x^k - x_*^k\right\|^2\right] - 2\gamma\left\langle\nabla f(x^k), x^k - x_*^k\right\rangle + \gamma^2\left\|\nabla f(x^k)\right\|^2 + \gamma^2\mathbb{E}_k\left[\left\|\nabla f(x^k;\xi^k) - \nabla f(x^k)\right\|^2\right]$$

$$\tag{27}$$

Since the function $f$ is $L$–smooth and $\nabla f(x_*^k) = 0$, we obtain

$$-2\gamma \left\langle \nabla f(x^k), x^k - x_*^k \right\rangle + \gamma^2 \left\| \nabla f(x^k) \right\|^2$$

$$= -2\gamma \left\langle \nabla f(x^k) - \nabla f(x_*), x^k - x_*^k \right\rangle + \gamma^2 \left\| \nabla f(x^k) - \nabla f(x_*) \right\|^2$$

$$\leq -2\gamma \left\langle \nabla f(x^k) - \nabla f(x_*), x^k - x_*^k \right\rangle + L\gamma^2 \left\langle \nabla f(x^k) - \nabla f(x_*), x^k - x_*^k \right\rangle$$

$$= \gamma \left( L\gamma - 2 \right) \left\langle \nabla f(x^k) - \nabla f(x_*), x^k - x_*^k \right\rangle.$$

Taking $\gamma \leq \frac{1}{L}$ and substituting the inequality to (27), we get

$$\mathbb{E}_k \left[ \left\| x^{k+1} - x_*^{k+1} \right\|^2 \right] \leq \mathbb{E}_k \left[ \left\| x^k - x_*^k \right\|^2 \right] - \gamma \left\langle \nabla f(x^k), x^k - x_*^k \right\rangle + \gamma^2 \mathbb{E}_k \left[ \left\| \nabla f(x^k; \xi^k) - \nabla f(x^k) \right\|^2 \right].$$

The $\sigma^2$–variance bounded ensures that

$$\mathbb{E}_k \left[ \left\| x^{k+1} - x_*^{k+1} \right\|^2 \right] \leq \mathbb{E}_k \left[ \left\| x^k - x_*^k \right\|^2 \right] - \gamma \left\langle \nabla f(x^k), x^k - x_*^k \right\rangle + \gamma^2 \sigma^2.$$

Due to convexity and Assumption 5.2, we can use Lemma C.2, which yields

$$\mathbb{E}_k \left[ \left\| x^{k+1} - x_*^{k+1} \right\|^2 \right] \leq \mathbb{E}_k \left[ \left\| x^k - x_*^k \right\|^2 \right] - \frac{\gamma\mu}{2} \left\| x^k - x_*^k \right\| + \gamma^2 \sigma^2$$

$$= \left( 1 - \frac{\gamma\mu}{2} \right) \mathbb{E}_k \left[ \left\| x^k - x_*^k \right\|^2 \right] + \gamma^2 \sigma^2.$$

Unrolling the recursion and taking the full expectation, we obtain

$$\mathbb{E} \left[ \left\| x^{k+1} - x_*^{k+1} \right\|^2 \right] \leq \left( 1 - \frac{\gamma\mu}{2} \right)^{k+1} \left\| x^0 - x_*^0 \right\|^2 + \frac{2\gamma\sigma^2}{\mu}$$

$\square$

## F  ASSUMPTIONS 5.1, 5.6 AND 5.7 IMPLY ASSUMPTION 5.2

**Theorem F.1.** *Let $\{f_i\}$ satisfy Assumption 5.1, 5.6, and Assumption 5.7 with constant $\mu$, then $f$ satisfies Assumption 5.2 with constant $\frac{\mu}{4}$.*

*Proof.* We fix $x \in \mathbb{R}^d$. Since Assumption 5.6 hold, then the functions share the closest solution $x_*$ to $x$. Assumption 5.7 ensures that

$$f_i(x) - f_i(x_*) \geq \frac{\mu}{2} \left\| x - x_* \right\|^2.$$

for all $i \in [n]$ (Karimi et al., 2016). Thus

$$f(x) - f(x_*) \geq \frac{\mu}{2} \left\| x - x_* \right\|^2.$$

Due to convexity, we get

$$f(x_*) \geq f(x) + \left\langle \nabla f(x), x_* - x \right\rangle.$$

Therefore

$$f(x) - f(x_*) \leq \left\langle \nabla f(x), x - x_* \right\rangle \leq \left\| \nabla f(x) \right\| \left\| x - x_* \right\| \leq \left\| \nabla f(x) \right\| \sqrt{\frac{2}{\mu}} \sqrt{f(x) - f(x_*)}$$

and

$$\frac{\mu}{4} \left( f(x) - f(x_*) \right) \leq \frac{1}{2} \left\| \nabla f(x) \right\|^2,$$

which is Assumption 5.2 with constant $\frac{\mu}{4}$.

$\square$

## G    FIRST-ORDER SIMILARITY AND SECOND-ORDER SIMILARITY

**Theorem 4.3.** *Consider the* Weighted SGD *method with* $f_i(x) \; : \; \mathbb{R} \; \to \; \mathbb{R}$ *such that* $f_i(x) = \beta \langle a_i, x \rangle + \frac{\beta}{2} \|x\|^2$, $a_i \in \mathbb{R}$ *for all* $i \in [n]$, $\frac{1}{n} \sum_{i=1}^{n} a_i = 0$, *and* $\|a_i\| = 1$, *where* $\beta > 0$ *is free parameter. Assume that there is no noise in the stochastic gradients, which means* $\nabla f_i(x; \xi_i) = \nabla f_i(x)$ *deterministically for all* $\xi_i \in \mathbb{S}_{\xi_i}$, $i \in [n]$, *and* $x \in \mathbb{R}$, *Then* Weighted SGD *converges to the point* $\frac{1}{n} \sum_{j=1}^{n} w_j B_j a_j$ *instead of* $x_* = 0$, *Assumption 4.1 (the first-order similarity) is satisfied with* $\delta_1 = 2\beta^2$, *and Assumption 4.2 (the second-order similarity) is satisfied with* $\delta_2 = 0$.

*Proof.* The first-order similarity of these functions is

$$\max_{x \in \mathbb{R}^d} \max_{i,j \in [n]} \|\nabla f_i(x) - \nabla f_j(x)\|^2 = \beta^2 \max_{i,j \in [n]} \|a_i - a_j\|^2 \leq 2\beta^2.$$

Thus, the parameter $\beta$ from the construction controls this similarity. Taking $\beta$ small, we increase similarity between the functions. At the same time, the distance $\left\| \frac{1}{n} \sum_{j=1}^{n} w_j B_j a_j \right\|$ between the minimum $x_* = 0$ and the point $\frac{1}{n} \sum_{j=1}^{n} w_j B_j a_j$ where Weighted SGD converges does not depend on $\beta$.

Notice that the second-order similarity between the functions is zero since $\nabla^2 f_i(x) = f_i''(x) = \beta$ for all $i \in [n]$.

The rest of the proof is almost the same as in Theorem 3.1. If $w_i B_i = 1$ for all $i \in [n]$, then Weighted SGD converges with $\gamma < \frac{2}{\beta}$ because Malenia SGD converges. If $w_i B_i \neq 1$ for all $i \in [n]$, then

$$x^{k+1} = x^k - \gamma \frac{1}{n} \sum_{i=1}^{n} w_i B_i \beta (a_i + x^k)$$

$$= \left( 1 - \gamma \frac{1}{n} \sum_{i=1}^{n} w_i B_i \beta \right) x^k - \gamma \beta \frac{1}{n} \sum_{i=1}^{n} w_i B_i a_i$$

$$= (1 - \gamma \beta) x^k - \gamma \beta \frac{1}{n} \sum_{i=1}^{n} w_i B_i a_i$$

$$= (1 - \gamma \beta)^{k+1} x^0 + \sum_{j=0}^{k} \gamma \beta (1 - \gamma \beta)^j \frac{1}{n} \sum_{i=1}^{n} w_i B_i a_i,$$

where the second equality due to the agreement $\frac{1}{n} \sum_{i=1}^{n} w_i B_i = 1$. If $\gamma \geq \frac{2}{\beta}$, then $x^{k+1}$ does not converge when $k \to \infty$. If $\gamma < \frac{2}{\beta}$, then

$$\lim_{k \to \infty} x^{k+1} = \frac{1}{n} \sum_{j=1}^{n} w_j B_j a_j.$$

$\square$

# H EXPERIMENTS

We conduct a comparison between Rennala SGD and Malenia SGD on both stochastic quadratic optimization tasks and real-world machine learning problems. These are standard quadratic optimization and computer vision problems, the design of which we explain in Section I. We developed a library that simulates the behavior of $n = 100$ workers. Both methods have two hyperparameters: step size $\gamma$ and parameter $S$. We do a grid search for both methods and find the best pairs in all setups. We start with synthetic quadratic optimization problems, which are generated *without and with the interpolation regime*. The procedure is described in Section I.1.

## H.1 WITHOUT INTERPOLATION

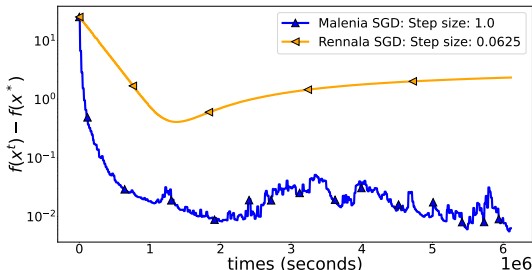

Figure 1: Comparison of the methods on a quadratic optimization problem *without interpolation*. We take the computation time $\tau_i = i^2$ for all $i \in [n]$.

In Figure 1, we present results without interpolation. The plots concur with the theory from Section 5, where we explain that it is essential to have interpolation to break the time complexity of Malenia SGD. Rennala SGD has biased gradient estimators and does not converge to a minimum of the quadratic optimization problem in Figure 1.

## H.2 WITH INTERPOLATION

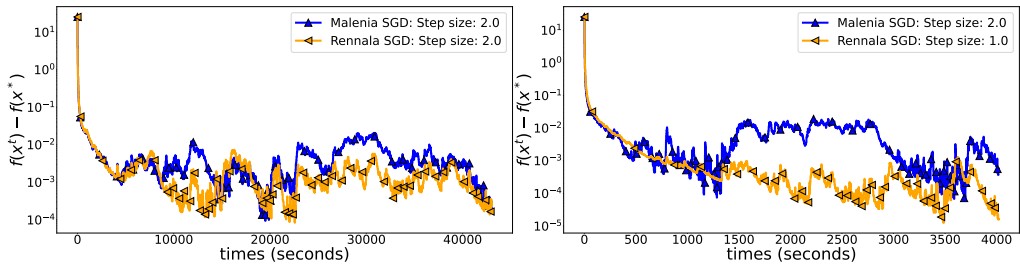

Figure 2: Comparison of the methods on quadratic optimization problems *with interpolation*. Times $\{\tau_i\}$ less diverse: *Left plot:* $\tau_i = \sqrt{i}$ for all $i \in [n]$. *Right plot:* $\tau_1 = 0.01, \tau_2 = 1, \ldots, \tau_n = 1$.

In Figures 2 and 3, we consider the methods in the interpolation regime. As expected, according to Section 5.2, Rennala SGD outperforms Malenia SGD in all experiments. We compare the methods with different $\{\tau_i\}$. In Figures 2, the times $\{\tau_i\}$ are less diverse, so the difference between the methods is less profound. In Figures 3, $\{\tau_i\}$ are more different; thus, we can see that Rennala SGD converges much faster to low function values because it has much less variance in the corresponding gradient estimator.

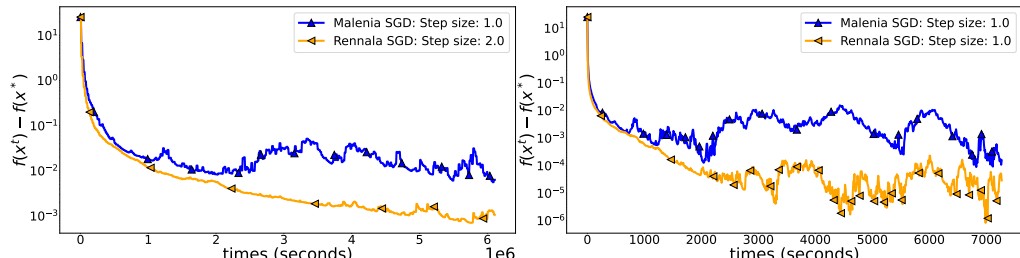

Figure 3: Comparison of the methods on quadratic optimization problems *with interpolation*. Times $\{\tau_i\}$ more diverse: *Left plot:* $\tau_i = i^2$ for all $i \in [n]$. *Right plot:* $\tau_1 = 0.001, \tau_2 = 1, \ldots, \tau_n = 1$.

### H.3 RESNET-18 AND CIFAR-10

We also verify how Rennala SGD and Malenia SGD work with ResNet-18 and the CIFAR-10 classification problem (Krizhevsky et al., 2009) (License: MIT). Both algorithms take step size $\gamma = 0.25$, sample a batch of size 128, and the smallest $S$ such that all workers calculate at least one batch. The dataset CIFAR-10 is split between the workers, so we consider the heterogeneous setting; all workers access different samples. The results of the experiments are presented in Figure 4. One can see that Rennala SGD converges faster in terms of accuracy, which might be explained by the fact that neural networks work in the interpolation regime.

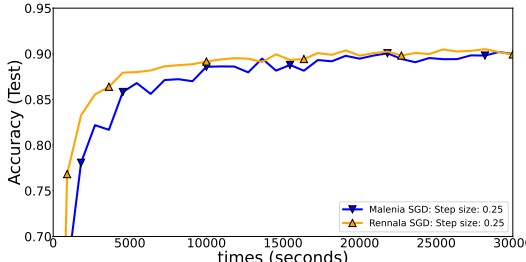

Figure 4: Comparison of the methods on the CIFAR-10 classification problem with ResNet-18. We take the computation time $\tau_i = i^2$.

## I EXPERIMENTS DETAILS

The experiments were run in Python 3 using an Intel(R) Xeon(R) Gold 6248 CPU @ 2.50GHz.

### I.1 QUADRATIC OPTIMIZATION TASK GENERATION PROCEDURE

In Section H, we perform experiments using synthetic quadratic optimization problems

$$\min_{x \in \mathbb{R}^d} \frac{1}{n} \sum_{i=1}^n \left( \frac{1}{2} x^\top \mathbf{A}_i x - x^\top b_i \right).$$

Below, we present the algorithm, based on (Szlendak et al., 2021), that generates these problems. In all experiments, we take $s = 3$ to ensure that the generated matrices are diverse. We take $n = 100$, $d = 100$, and $\lambda = 0.001$. The stochastic gradients are equal to the true gradients plus standard Gaussian noise added to the coordinates to emulate stochasticity.

With these parameters and procedures, we run the experiments from Section H.1. To conduct the experiments from Section H.2 in the interpolation regime, we take the matrices $\mathbf{A}_1, \cdots, \mathbf{A}_n$, vectors $b_1, \cdots, b_n$ returned by Algorithm 3. Let $\bar{x}_*$ be the solution of the quadratic optimization problem $\frac{1}{n} \sum_{i=1}^n \mathbf{A}_i \bar{x}_* = \frac{1}{n} \sum_{i=1}^n b_i$. Then, we redefine the vectors $\{b_i\}$ as $b_i = \mathbf{A}_i \bar{x}_*$ to ensure that we are working in the interpolation regime. With this strategy, the matrices are still different, and the functions $\{f_i\}$ are not equal.

---

**Algorithm 3** Generate quadratic optimization tasks

---

1: **Parameters:** number nodes $n$, dimension $d$, regularizer $\lambda$, and noise scale $s$.
2: **for** $i = 1, \ldots, n$ **do**
3:     Generate random noises $\eta_i^s = 1 + s\zeta_i^s$ and $\eta_i^b = s\zeta_i^b$, i.i.d. $\zeta_i^s, \zeta_i^b \sim \mathcal{N}(0, 1)$
4:     Take vector $b_i = \frac{\eta_i^s}{4}(-1 + \eta_i^b, 0, \cdots, 0) \in \mathbb{R}^d$
5:     Take the initial tridiagonal matrix

$$\mathbf{A}_i = \frac{\eta_i^s}{4} \begin{pmatrix} 2 & -1 & & 0 \\ -1 & \ddots & \ddots & \\ & \ddots & \ddots & -1 \\ 0 & & -1 & 2 \end{pmatrix} \in \mathbb{R}^{d \times d}$$

6: **end for**
7: Take the mean of matrices $\mathbf{A} = \frac{1}{n} \sum_{i=1}^n \mathbf{A}_i$
8: Find the minimum eigenvalue $\lambda_{\min}(\mathbf{A})$
9: **for** $i = 1, \ldots, n$ **do**
10:     Update matrix $\mathbf{A}_i = \mathbf{A}_i + (\lambda - \lambda_{\min}(\mathbf{A}))\mathbf{I}$
11: **end for**
12: Take starting point $x^0 = (\sqrt{d}, 0, \cdots, 0)$
13: **Output:** matrices $\mathbf{A}_1, \cdots, \mathbf{A}_n$, vectors $b_1, \cdots, b_n$, starting point $x^0$

---

### I.2 EXPERIMENTS WITH RESNET AND CIFAR-10

In Section H.3, we consider the standard computer vision classification problem with ResNet-18 (He et al., 2016) and CIFAR-10 (Krizhevsky et al., 2009). We conduct the experiments using PyTorch and implement both Rennala SGD and Malenia SGD optimizers. For reproducibility, we use the default ResNet-18 architecture provided in PyTorch and split randomly and evenly the CIFAR-10 dataset across multiple workers to create a heterogeneous data distribution scenario. We use standard preprocessing techniques for CIFAR-10, including normalization and random cropping, and train the network for a fixed number of epochs. The performance metrics include top-1 accuracy. In total, we solve the optimization problem

$$\min_{x \in \mathbb{R}^d} \frac{1}{n} \sum_{i=1}^n \left( \frac{1}{m} \sum_{j=1}^m \text{loss}(\text{ResNet}(a_{ij}; x), y_{ij}) \right),$$

where "loss" is the standard cross-entropy loss, $\{a_{ij}, y_{ij}\}$ are samples from CIFAR-10 splitted between the workers.

