# OpenReview forum: "Bridging the Gap Between Homogeneous and Heterogeneous Asynchronous Optimization is Surprisingly Difficult"
_ICLR.cc/2026/Conference — Submitted to ICLR 2026_

### Official Review · Reviewer_SYHQ · 2025-10-30

**Soundness:** 4
**Presentation:** 3
**Contribution:** 2
**Rating:** 4
**Confidence:** 4

**Summary:**

This paper explores the gap between time complexities under homogeneous and heterogeneous worker losses for a  class of asynchronous optimization algorithms. These methods distribute the gradient computations across workers until “a sufficient number” of gradients have been computed and then update the iterate, but they differ in how the next iterate is computed. In this paper, the authors investigate under which assumptions one can close the gap between the lower bounds for homogeneous and heterogeneous worker losses.

I find the paper interesting and important when it comes to expanding our theoretical understanding of the time complexities of the particular classes of methods under consideration. However, the scope is limited (weighted gradient methods) and the analysis does not make any advances on the algorithmic side (it simply finds conditions under which it is possible to get the desired compute-time dependence, relying on a mixing already been proposed in the literature).

**Strengths:**

1. The authors address a well-defined question (under what assumption can a weighted gradient method of the type studied in the paper attain optimal worker time dependence thus far only associated with homogeneous worker losses) and present a sound a convincing argument with constructive “difficult functions” for each considered scenario that shows that the lower bounds cannot be attained.
2. The overview of results in the area is comprehensive and clear.
3. The final result is clear: under a set of strong conditions, Rennala SGD attains the desired  compute time dependence, and no other weighting can improve its performance.

**Weaknesses:**

The paper would have been easier to read with at least the very basic notation (e.g. \tilde{\Theta}) in the main document. The fact that the author frames a lot of the discussion around Renala vs Malania SGD, instead of the underlying ideas of the methods (equally weighted gradients vs importance sampled gradients/unbiased worker gradient estiamtes) makes it more difficult to keep up a good intuition while reading the paper.

I am also not fully convinced by the framing of the paper. It would be good if the authors could provide more intuition about what they are trying to do and why? Because now I come away from reading the paper with the feeling that this is more of a “robustness analysis” of Rennala SGD (how much can we wiggle the assumptions from homogeneity and still get the same compute time dependence in the convergence result). Clearly, this is not the way the authors wants us to read the paper (cf. the very last paragraph of Section 1), but I think they need to make a slightly stronger argument for why their result is more than this. For example, it is clear that they want to attain the harmonic mean dependence on worker compute times, but why this particular form? Presumably, you would like to avoid waiting for the slowest worker, while being able to use very fast compute nodes to reduce variance and progress to the next iterate. But is there something fundamental with this dependence? Second, in general, there should be little hope of not waiting for the slowest worker, unless it is not needed (strong interpolation), so the results are not very surprising. It is not clear to me why the authors were hoping that gradient similarity or weak interpolation would help.

**Questions:**

1. The class of methods is rather limited. What is the reason to only consider weighted SGD and no other techniques?
2. Can you elaborate on what is fundamental with the  harmonic mean compute-time dependence in the heterogeneous setting?
3. Can you elaborate on why you would expect that gradient similarlity or weak interpolation would help?
4. In your model for heterogeneity, you assume that the variances of worker subgradients are identical. When is that reasonable in practical ML settings? What would happen if you allowed them to be different?
5. For the simulations, I could not find how you generated the \tau_i’s, and how they vary in magnitudes and time (you emphasize in the introduction that your setting includes non-constant computation times)
6. It seems that Theorem E.4 is new. Is that true? Why was it relegated to an appendix?

---

> ### Author Response · Authors · 2025-11-16
> **Official Comment by Authors (Part 1 / 3)**
>
> Thank you for the comments and questions!
>
> > The paper would have been easier to read with at least the very basic notation (e.g. \tilde{\Theta}) in the main document.
>
> The notations section was in the appendix, but we've just moved it to the main part in the updated PDF.
>
> > The fact that the author frames a lot of the discussion around Renala vs Malania SGD, instead of the underlying ideas of the methods (equally weighted gradients vs importance sampled gradients/unbiased worker gradient estiamtes) makes it more difficult to keep up a good intuition while reading the paper.
>
> Rennala SGD and Malenia SGD are described from different angles in Sections 2, 1.2, and Algorithm 1. They are simply vanilla SGD methods with asynchronous collections of stochastic gradients and different averaging schemes. If you point out the part that is unclear, we would be happy to clarify it.
>
> > I am also not fully convinced by the framing of the paper.
>
> Let us explain this part. We know that the dependence
> $\Theta((\frac{1}{n} \sum_{i=1}^{n} \tau_{i}))$
> on $\{\tau_i\}$ is optimal in the *heterogeneous* setting and cannot be improved under standard assumptions (L-smothness (Assumption 1.1), bounded variance (Assumption 1.2), convexity (Assumption 5.1)). The dependence $\Theta((\frac{1}{n} \sum_{i=1}^{n} \tau_{i}))$ is arguably "bad" because it linearly depends on $\tau_i$. For instance, if we take $\tau_n \to \infty$, then the complexity also tends to $\infty$. In the *homogeneous* setting, the dependence $\Theta((\frac{1}{m} \sum_{i=1}^{n} \frac{1}{\tau_{i}})^{-1} (\cdot))$ is harmonic-like, which is much better, and, for example, if $\tau_n \to \infty$, then the complexity remains stable, as we explain in Section 1.2 (see "Difference between the two settings").
>
> > It would be good if the authors could provide more intuition about what they are trying to do and why?
>
> *What are we trying to do?* We are trying to understand under which assumptions we can improve the dependence $\Theta((\frac{1}{n} \sum_{i=1}^{n} \tau_{i})).$ We want to find a set of assumptions as weak as possible that would allow us to improve the dependence $\Theta((\frac{1}{n} \sum_{i=1}^{n} \tau_{i}))$ and get closer to $\Theta((\frac{1}{m} \sum_{i=1}^{n} \frac{1}{\tau_{i}})^{-1} (\cdot))$ (not necessarily equal).
>
> *Why?* Because we want to improve the pessimistic dependence $\Theta((\frac{1}{n} \sum_{i=1}^{n} \tau_{i}))$. Suppose neither you nor we know the results of this paper in advance. It might be possible that adding a very weak assumption on the functions $f_i$ is enough to improve the complexity. If we can find such an assumption, then we could begin adjusting existing architectures or redesigning the data allocation so that this assumption holds, leading the acceleration. This assumption would be very useful in understanding how to speed up training.
>
> Unfortunately, based on the results of this paper, it seems that such a “very weak assumption” does not exist, and we must assume arguably strong assumptions to improve the dependence $\Theta((\frac{1}{n} \sum_{i=1}^{n} \tau_{i})).$ We believe that this exploratory research is an important contribution.

---

> > ### Author Response · Authors · 2025-11-16
> > **Official Comment by Authors (Part 2 / 3)**
> >
> > > Because now I come away from reading the paper with the feeling that this is more of a “robustness analysis” of Rennala SGD (how much can we wiggle the assumptions from homogeneity and still get the same compute time dependence in the convergence result).
> >
> > This is also a valid point of view. Indeed, another equivalent research view is to start with the homogeneous setting and understand which assumptions can be relaxed. One would obtain the same conclusions as in our approach, where we start with the heterogeneous setting and try to add assumptions that are as weak as possible.
> >
> > > Clearly, this is not the way the authors wants us to read the paper (cf. the very last paragraph of Section 1), but I think they need to make a slightly stronger argument for why their result is more than this. For example, it is clear that they want to attain the harmonic mean dependence on worker compute times, but why this particular form?
> >
> > > Can you elaborate on what is fundamental with the harmonic mean compute-time dependence in the heterogeneous setting?
> >
> > Not necessarily the harmonic mean dependence. We would be happy with any non-marginal improvement of $\Theta((\frac{1}{n} \sum_{i=1}^{n} \tau_{i}))$. The harmonic mean is our ultimate goal. If a method achieves a harmonic mean dependence, then one slow worker does not ruin the time complexity. In contrast, the arithmetic mean dependence is non-robust: one large $\tau_i$ slows the entire method. The robustness property of the harmonic mean dependence makes it fundamental because it is the ideal behavior we would want from an asynchronous method.
> >
> > > Presumably, you would like to avoid waiting for the slowest worker, while being able to use very fast compute nodes to reduce variance and progress to the next iterate. But is there something fundamental with this dependence?
> >
> > Of course, this is a fundamentally practical task: to understand when it is possible to develop a method that is robust to slow computations. Clearly, $\Theta((\frac{1}{n} \sum_{i=1}^{n} \tau_{i}))$ is non-robust. In real distributed and federated learning settings, the computation and communication times are noisy.
> >
> > > Second, in general, there should be little hope of not waiting for the slowest worker, unless it is not needed (strong interpolation), so the results are not very surprising. It is not clear to me why the authors were hoping that gradient similarity or weak interpolation would help.
> >
> > > Can you elaborate on why you would expect that gradient similarlity or weak interpolation would help?
> >
> > We can’t simply state that “there should be little hope.” This isn’t a guessing game. Someone actually needs to sit down and carefully examine whether such an improvement is possible or not. Our goal was precisely to investigate this question in a research setting.
> >
> > Gradient similarity, Hessian similarity, and weak interpolation are natural assumptions that have helped improve convergence in many related optimization settings and problems (e.g., [3,4,5]). In [4], gradient and Hessian similarities helped show that Local SGD (one of the most popular methods) can achieve smaller iteration complexity. In [5], the authors also show that the communication complexity can be reduced if the Hessian similarity is small. It is also well known that weak interpolation allows one to prove the convergence of SGD exactly to the minimum (without a neighborhood) (e.g., [6]).
> >
> > Therefore, it was reasonable to explore whether these assumptions could also mitigate the dependence on the slowest workers here.
> >
> > > The class of methods is rather limited. What is the reason to only consider weighted SGD and no other techniques?
> >
> > It is reasonable to consider Weighted SGD, a "meta" method or a family of methods, since it naturally generalizes both Rennala SGD and Malenia SGD. It is also reasonable to expect that if Malenia SGD can be improved, then the improved method would fall within the Weighted SGD family. There are several important works that analyze families of methods, with meaningful implications for the optimization community (for example, [1] studies a subclass of Asynchronous SGD, and [2] focuses on a single method, Local SGD). We acknowledge that this is a limitation of the work (and we have added this limitation to the conclusion section), but we believe this limitation is minor.

---

> > > ### Author Response · Authors · 2025-11-16
> > > **Official Comment by Authors (Part 3 / 3)**
> > >
> > > > In your model for heterogeneity, you assume that the variances of worker subgradients are identical. When is that reasonable in practical ML settings? What would happen if you allowed them to be different?
> > >
> > > The consequences and the theory would be the same, except that the formulas would be more bulky. For instance, let consider the homogeneous setting where workers have access to the stochastic gradients but with different variances:
> > >
> > > $$E||\frac{1}{\sum_{i=1}^n B_i^k} \sum\limits_{i=1}^{n} \sum\limits_{j=1}^{B_i^k} \nabla f(x^k;\xi^k_{ij}) - \nabla f(x^k)||^2 \leq \frac{1}{(\sum_{i=1}^n B_i^k)^2} \sum\limits_{i=1}^{n} B_i^k \sigma^2_i.$$
> > >
> > > Similarly to the case when the variances are equal, in Rennala SGD, we wait for the moment when $\frac{1}{(\sum_{i=1}^n B_i^k)^2} \sum\limits_{i=1}^{n} B_i^k \sigma^2_i \leq \varepsilon,$ which takes at most
> > > $t = \frac{\sum_{i=1}^{n} \sigma^2_i / \tau_i}{(\sum_{i=1}^n 1 / \tau_i)^2}$
> > > seconds. When $\sigma_i = \sigma,$ we get the harmonic dependence, but with different $\sigma_i$ the dependence is slightly more complex. The same derivations can be done in the heterogeneous setting. The gap between the settings would be similar to the case when $\sigma_i = \sigma.$
> > >
> > > > For the simulations, I could not find how you generated the \tau_i’s, and how they vary in magnitudes and time (you emphasize in the introduction that your setting includes non-constant computation times)
> > >
> > > In all experiments, we fix $\tau_i$, which is done to verify the theoretical results from the main text. The values of ${\tau_i}$ are listed under the figures (please consider the updated PDF, where the values of $\tau_i$ are presented).
> > >
> > > Yes, our theory supports the non-constant computation times described in Section A.
> > >
> > > > It seems that Theorem E.4 is new. Is that true? Why was it relegated to an appendix?
> > >
> > > Yes, Theorem E.4 is new. We decided to place it in the appendix to keep the reader’s attention focused on the main research questions and contributions described in Section 1.2.
> > >
> > > ---
> > >
> > > **Thank you once again! We believe that we have addressed all the questions and comments, and we hope that the reviewer will reconsider the score. If you have more questions, please let us know.**
> > >
> > > [1]: S. Vaswani, A. Mishkin, I. Laradji, Mark Schmidt, Gauthier Gidel, and Simon Lacoste-Julien. Painless stochastic gradient: Interpolation, line-search, and convergence rates. NeurIPS 2019.
> > >
> > > [2]: G. Garrigos and R. M. Gower. “Handbook of Convergence Theorems for (Stochastic) Gradient Methods.” (2023).
> > >
> > > [3]: R. Szlendak, A. Tyurin, P. Richtárik Permutation compressors for provably faster distributed nonconvex optimization (ICLR 2022)
> > >
> > > [4]: R. Luo, S. Stich, S. Horváth, M. Takáč Revisiting LocalSGD and SCAFFOLD: Improved Rates and Missing Analysis (AISTASTS 2025)
> > >
> > > [5]: Y. Takezawa, X. Jiang, A. Rodomanov, S. Stich Exploiting Similarity for Computation and Communication-Efficient Decentralized Optimization (ICML 2025)
> > >
> > > [6]: E. Gorbunov, F. Hanzely, P. Richtárik A unified theory of SGD: Variance reduction, sampling, quantization and coordinate descent (AISTATS 2020)

---

> > > > ### Author Response · Authors · 2025-11-27
> > > >
> > > > Dear Reviewer,
> > > >
> > > > We hope that you have had time to review our response and that you found your questions and comments fully addressed. We would be glad to clarify any further questions.
> > > >
> > > > Thank you for your time!
> > > >
> > > > Authors

---

### Official Review · Reviewer_mKAf · 2025-10-31

**Soundness:** 3
**Presentation:** 2
**Contribution:** 3
**Rating:** 6
**Confidence:** 4

**Summary:**

This paper considers the problem of distributed optimization where n workers/clients are working in parallel asynchronously to optimize an objective function and find the minima, drawing motivation from distributed and federated ML. Each worker has their own function f_i, dataset D_i, and are able to compute their local stochastic gradient on f_i(x,D_i). The global objective is a sum of the expected values of the local objectives. The paper specifically revisit the mathematical analysis of asynchronous SGD under the "heterogenous" case, i.e., where each client has a different data distribution, attempting to improve the time complexity under alternate mathematical assumptions.
They first show that the existing pessimistic bounds are indeed tight and cannot be improved under certain classes of assumptions considered. Then finally, they introduce an alternate assumption under which the guarantees for the "heterogenous" case matches the "homogenous" case. Primarily a theoretical paper with several takeaways, no experiments.

**Strengths:**

-- The paper first introduces a weighted SGD as a general way of writing the distributed SGD solution. The two specific cases Rennala SGD (optimal in homogenous) and Malenia SGD (optimal in heterogenous) come out as special cases of the general weighted SGD solution. This is helpful since now the weighted SGD can be directly analyzed.

-- First, they show that convergence if Malenia SGD is challenging. Even first order or second order similarity between the functions across all the clients do not necessarily help. I greatly appreciate this detailed analysis of a negative result, highlighting the mathematical insights on why it doesn't work.

-- Then they consider an alternate set of interpolation assumptions under which it is possible to improve the time complexity of Malenia SGD.

-- The takeaways of the theoretical results are quite nice and insightful.

**Weaknesses:**

-- It is not clear how the new set of assumptions compare with the previous assumptions. Which assumptions are weaker or stronger? Is it possible to give intuition on what these new assumptions mean, perhaps using a toy example on the functions f_i?

-- For instance, are there toy functions/ classes of functions that satisfy the old assumptions but not the new? Vice versa, are there functions that satisfy the new assumptions but not the old?

-- Theorem 3.1 gradient condition: Do you need an expectation $\Delta f(x;\xi)=\Delta f(x)$

-- The table on page 7 with the contributions is nice. But the first column can be clearer about homogenous/heterogenous case to make the contributions clearer.

-- Missed several related works on asynchronous SGD
[1] Xiangru Lian, Yijun Huang, Yuncheng Li, and Ji Liu. Asynchronous parallel stochastic gradient for nonconvex optimization. In Advances in Neural Information Processing Systems, pages 2737–2745, 2015.
[2] Wei Zhang, Suyog Gupta, Xiangru Lian, and Ji Liu. Staleness-aware async-sgd for distributed deep learning. In International Joint Conference on Artificial Intelligence, pages 2350–2356. AAAI Press, 2016.
[3] Dutta, S., Joshi, G., Ghosh, S., Dube, P., & Nagpurkar, P. Slow and stale gradients can win the race: Error-runtime trade-offs in distributed SGD. In International conference on artificial intelligence and statistics (pp. 803-812) 2018.

-- Also see the citations therein

-- Abstract is too domain-oriented with specific jargon and is not very accessible for a broader audience.

-- Though the contribution is theoretical, and experimental results are not necessary - But I was curious if an asynchronous algorithm is implemented on say a strongly convex loss function or one that satisfies the necessary assumptions, how well do these bounds align with practice? Are there empirical studies on Rennala SGD (optimal in homogenous) and Malenia SGD (optimal in heterogenous) and how good do the time complexity bounds align with practice?

**Questions:**

Could you compare the two sets of assumptions, and what do they mean intuitively for real functions?

Is one stronger or weaker?

What are some toy functions that satisfy one but not the other?

**Details Of Ethics Concerns:**

Not needed

---

> ### Author Response · Authors · 2025-11-16
>
> Thank you for the review!
>
> > It is not clear how the new set of assumptions compare with the previous assumptions. Which assumptions are weaker or stronger? Is it possible to give intuition on what these new assumptions mean, perhaps using a toy example on the functions f_i?
>
> > Could you compare the two sets of assumptions, and what do they mean intuitively for real functions? Is one stronger or weaker? What are some toy functions that satisfy one but not the other?
>
> > For instance, are there toy functions/ classes of functions that satisfy the old assumptions but not the new? Vice versa, are there functions that satisfy the new assumptions but not the old?
>
> Assumptions 4.1 and 4.2 are standard assumptions that capture the similarity of functions through gradients and Hessians.  For instance, in the case of Assumption 4.1, consider the example $f_i(x) = \langle a_i, x \rangle$ for which
> $$||\nabla f_i(x) - \nabla f_j(x)|| = ||a_i - a_j||.$$
>
> The first non-standard assumptions that might not be clear are Assumption 5.4 (Weak Interpolation) and Assumption 5.6 (Strong Interpolation). In Theorem 3.1, we provide an example in which neither assumption is satisfied. We now present an example where Assumption 5.4 is satisfied, but Assumption 5.6 is not:
>
> Take $x=(x_1,x_2)\in R^2$ and define two quadratic functions
> $$
> f_1(x)=\frac{1}{2}x_1^2,\qquad
> f_2(x)=\frac{1}{2}x_2^2,\qquad
> f(x)=\frac{1}{2}\bigl(f_1(x)+f_2(x)\bigr)=\frac{1}{4}(x_1^2+x_2^2).
> $$
> The function $f$ has the unique minimizer $x^*=(0,0)$, which is also a minimizer of each $f_i$, so weak
> interpolation holds. However, strong interpolation fails: for $\tilde{x}=(0,1)$ we have $f_1(\tilde{x})=0$
> (i.e., $\tilde{x}$ is a minimizer of $f_1$), while $f(\tilde{x})=\frac{1}{4}>0$, so $\tilde{x}$ is not a minimizer of $f$.
>
> The last assumption that we introduce is Assumption 5.7. Compared to Assumption 5.2, Assumption 5.7 requires *each function* to satisfy the Polyak-Łojasiewicz condition (generalization of strong convexity). One particular example when Assumption 5.2 is satisfied, and Assumption 5.7 is not is $f_1(x) = 0.5 \delta x^2$ and $f_2(x) = \mu x^2,$ where $\delta$ is a small constant ($\delta \ll \mu$). $f_1$ does not satisfy Assumption 5.7 with parameter $\mu$ when $\delta$ is very small, but their average satisfy Assumption 5.2 with parameter $\mu$.
>
> If you need any extra clarifications, then let us know.
>
> > Theorem 3.1 gradient condition: Do you need an expectation $\nabla f(x;\xi)=\nabla f(x)$
>
> This construction is deterministic and non-random, so $E[\nabla f(x;\xi)] = \nabla f(x;\xi) = \nabla f(x).$
>
> > The table on page 7 with the contributions is nice. But the first column can be clearer about homogenous/heterogenous case to make the contributions clearer.
>
> Thank you. The table compares methods in the fully heterogeneous setting and lists the extra assumptions the methods require to work. We've clarified this in the updated PDF.
>
> > Missed several related works on asynchronous SGD ...
>
> Thank you. We’ve added the missing references to the updated PDF.
>
> > Though the contribution is theoretical, and experimental results are not necessary ...
>
> Please consider our Section I with experiments.
>
> ---
>
> Thank you for the comments and questions. If you have any other questions, then let us know.

---

### Official Review · Reviewer_Mpme · 2025-11-03

**Soundness:** 3
**Presentation:** 4
**Contribution:** 1
**Rating:** 4
**Confidence:** 3

**Summary:**

In this work, the authors study the complexity of heterogeneous asynchronous optimization. In particular, given workers with non-constant computation time of gradients that is bounded for worker $i$ by $\tau_i$, the authors investigate the optimal complexity achievable with an asynchronous method that they call Weighted SGD. In particular, two previous methods for asynchronous optimization known as Rennala SGD and Malenia SGD are special cases of Weighted SGD. The authors prove a series of negative results for this method under various combinations of function/data similary/interpolation and convexity/PL-condition as well as smoothness.

**Strengths:**

1. The work is presented really well. I found it very accessible and in fact an enjoyable read. The main takeaway messages are nicely emphasized and all assumptions are clearly stated.
2. Asynchronous optimization is a sensible approach to distributed systems, especially when hardware failures are possible, which can be very important given the never ending growth of large-scale clusters.

**Weaknesses:**

1. I'd say the main weakness of the paper is the significance. The studided methods were proposed and analyzed in prior literature. The non-convergence result is mostly trivial. Moreover, the results for first-order and second-order similarity are only applicable to Weighted SGD, which in itself is not a particularly interesting method.

2. I'm also concerned about how the results for the similary are presented. The authors wrote:
> One might expect that when both $\delta\_1$ or $\delta\_2$ are small, it would be possible to exploit the similarity and design a method with smaller variance and better dependence on $\{\tau\_i\}$. Surprisingly, it is not the case

But the authors don't actually show that no other method exists in this setting, they only present a counterexample to Weighted SGD with fixed weights. I find this kind of presentation rather misleading, in particular the way it is written in Takeaway 3 and Takeaway 4.

3. I also feel the topic of the paper is drifting from the practical methods. Malenia SGD requires each worker to submit at least one gradient, which is not something we would use in practice when a node failure is possible. It makes me question if the methods presented in this paper are truly possibly useful. I suspect that in practice it is much better to proceed without weighting for all nodes to submit at least a gradient.

**Questions:**

1. Do the authors believe Malenia SGD is a practical method? What happens if there is a failure?

2. More generally, what kind of setting are the authors motivated by? It doesn't appear that we have data heterogeneity in distributed computation on clusters as we can simply duplicate the data.

---

> ### Author Response · Authors · 2025-11-16
> **Official Comment by Authors (Part 1)**
>
> Thank you for your review.
>
> > I'd say the main weakness of the paper is the significance. The studided methods were proposed and analyzed in prior literature. The non-convergence result is mostly trivial. Moreover, the results for first-order and second-order similarity are only applicable to Weighted SGD, which in itself is not a particularly interesting method.
>
> We believe that the significance of our contribution lies in this exploration process. While previous work noted the existence of the gap, our contribution goes further by systematically investigating which assumptions are sufficient and which are insufficient to eliminate it. Analyzing the gap between existing algorithms is as important as developing new ones.
>
> There are hundreds of papers that analyze the classical SGD method, which was developed around 75 years ago, and the community still appreciates research on SGD. Similarly, we analyze the existing optimal Rennala SGD and Malenia SGD methods in the corresponding settings and want to understand how we can reduce the gap between them, which is an important task to understand the difference between the homogeneous and heterogeneous settings.
>
> It is reasonable to consider Weighted SGD, a “meta” method or a family of methods, since it naturally generalizes both the Rennala SGD and Malenia SGD methods. It is also reasonable to assume that if it is possible to improve Malenia SGD, then such a method would belong to the Weighted SGD family. There are many important works that analyze families of methods, with significant implications for the optimization community (e.g., [1] considers a subclass of Asynchronous SGD, and [2] focuses on a single method, Local SGD).
>
> > The non-convergence result is mostly trivial.
>
> We respectfully disagree that the non-convergence result is mostly trivial. To establish this result, we developed very delicate, non-trivial, and genuinely new constructions, which were not considered before. We can’t simply state that “is mostly trivial.” Someone actually needs to sit down and carefully examine whether such an improvement is possible or not. Our goal was precisely to investigate this question in a research setting.
>
> There are many works where the gradients and Hessian similarity and weak interpolation helped [3,4,5,6]. For instance, In [4], gradient and Hessian similarities allowed showing that the complexity of Local SGD can be improved.
>
> Thus, it was reasonable to investigate whether these assumptions could also help in our setting.
>
> > I'm also concerned about how the results for the similary are presented. The authors wrote ...
>
> Yes, Takeaway 3 and Takeaway 4 might be too strong. Please consider the updated PDF, where we now emphasize that the takeaways hold only for the family of methods Weighted SGD. As we explain in Section 2, Weighted SGD is a natural generalization of the two optimal methods. We acknowledge that this is a limitation of the work (and we have added this limitation to the conclusion section), but we believe this limitation is minor.

---

> ### Author Response · Authors · 2025-11-16
> **Official Comment by Authors (Part 2)**
>
> > More generally, what kind of setting are the authors motivated by? It doesn't appear that we have data heterogeneity in distributed computation on clusters as we can simply duplicate the data.
>
> The heterogeneous setting (when data cannot be duplicated) is very important in the community. The whole field of federated learning was developed to analyze this scenario. Due to privacy and other regulatory constraints, it is often infeasible to collect all data in one place in practice. We consider both the homogeneous and heterogeneous setups, and the goal of this work is to identify theoretical assumptions that are as weak as possible while still enabling improvements for asynchronous methods in heterogeneous settings. In other words, if data cannot be duplicated, what assumptions on the data or the function are needed in order to train as fast as if the data were duplicated? This question is important.
>
> > I also feel the topic of the paper is drifting from the practical methods. Malenia SGD requires each worker to submit at least one gradient, which is not something we would use in practice when a node failure is possible. It makes me question if the methods presented in this paper are truly possibly useful. I suspect that in practice it is much better to proceed without weighting for all nodes to submit at least a gradient.
>
> > Do the authors believe Malenia SGD is a practical method? What happens if there is a failure?
>
> As we explain in the previous comment, the topic and methods are important and practical. Yes, Malenia SGD requires each worker to submit at least one gradient. But from the point of view of time complexity, it is necessary to do due to the lower bounds.
>
> Of course, one can develop a method in which some workers do not submit a gradient in every iteration. However, this would lead to slower convergence or even divergence. This is not what we analyze in the paper. Instead, we take the fastest method, Malenia SGD, and try to understand whether it is possible to make it even faster.
>
> ---
>
> **Thank you once again. We believe that we have addressed all the comments. In the updated PDF, all remaining confusion has been resolved. If you have more questions, please let us know.**
>
> [1]: Y. Arjevani, O. Shamir, N. Srebro, A tight convergence analysis for stochastic gradient descent with delayed updates (ALT 2020)
>
> [2]: MR. Glasgow, H. Yuan, T. Ma, Sharp bounds for federated averaging (local sgd) and continuous perspective (AISTATS 2022)
>
> [3]: R. Szlendak, A. Tyurin, P. Richtárik Permutation compressors for provably faster distributed nonconvex optimization (ICLR 2022)
>
> [4]: R. Luo, S. Stich, S. Horváth, M. Takáč Revisiting LocalSGD and SCAFFOLD: Improved Rates and Missing Analysis (AISTASTS 2025)
>
> [5]: Y. Takezawa, X. Jiang, A. Rodomanov, S. Stich Exploiting Similarity for Computation and Communication-Efficient Decentralized Optimization (ICML 2025)
>
> [6]: E. Gorbunov, F. Hanzely, P. Richtárik A unified theory of SGD: Variance reduction, sampling, quantization and coordinate descent (AISTATS 2020)

---

> > ### Comment · Reviewer_Mpme · 2025-11-17
> >
> > I thank the authors for their response.
> >
> > > It is also reasonable to assume that if it is possible to improve Malenia SGD, then such a method would belong to the Weighted SGD family.
> >
> > Why is it reasonable? If it's obvious to the authors then it should be proven and it would make the results much stronger. Otherwise it's just a guess and it doesn't exclude the possibilty of a better method.
> >
> > > To establish this result, we developed very delicate, non-trivial, and genuinely new constructions, which were not considered before. We can’t simply state that “is mostly trivial.”
> >
> > The constructed function in the proof of Theorem 5.8 is a 2-dimensional quadratic with additive noise, which is not new at all. It might be tedious to analyze it precisely, but the construction itself is rather straightforward.
> >
> > > The whole field of federated learning was developed to analyze this scenario.
> >
> > The FL setting would indeed by of interest but as far as I can see the requirement for each worker to keep sending updates makes the studied methods inapplicable in that setting. If you're implying that asynchronous FL is impossible without this, it suggests that something is missing in the way the problem is formulated. A more reasonable setup would be if the workers are sampled and are not expected to communicate more than a single update (though potentially with many gradients).

---

> ### Author Response · Authors · 2025-11-17
>
> Thank you for your quick response.
>
> > Why is it reasonable? If it's obvious to the authors then it should be proven and it would make the results much stronger. Otherwise it's just a guess and it doesn't exclude the possibilty of a better method.
>
> Our motivation comes from the fact that both Rennala SGD and Malenia SGD are instances of Weighted SGD:
>
> $$x^{k+1} = x^{k} - \gamma g^k_{w}, \quad g^k_{w} = \frac{1}{n} \sum\limits_{i=1}^{n} w_i^k \sum\limits_{j=1}^{B_i^k} \nabla f_i(x^k;\xi^k_{ij})$$
>
> If we take $w_i^k = \frac{n}{\sum_{i=1}^n B_i^k}$ for all $i \in [n],$ we get Rennala SGD. If we take $w_i^k = \frac{1}{B_i^k},$ we get Malenia SGD. In this work, we analyze whether it is possible to find another instance of Weighted SGD with better guarantees.
>
> ---
>
> There are many works that analyze a particular method or a subfamily of methods. For instance, [2] analyzes only SGD and Adagrad, and that is it. [6] only focuses on the delayed SGD method. Such works are important for the community.
>
> ---
>
> > The constructed function in the proof of Theorem 5.8 is a 2-dimensional quadratic with additive noise, which is not new at all. It might be tedious to analyze it precisely, but the construction itself is rather straightforward.
>
> There is nothing wrong with using quadratic functions in the construction. Many modern works use quadratic functions in their lower bounds (e.g., [1]).
>
> The reviewer says that “the construction itself is rather straightforward.” It is difficult to respond to such a subjective comment, but we will try to explain why the construction is not trivial. There are several important and non-trivial details in designing the coefficients, such as choosing the correct coefficients of the quadratic functions.
> For instance, the coefficient $\frac{\frac{\mu}{B_i}}{\frac{2}{n} \sum_{i=1}^n \frac{1}{B_i}}$ in the proof of Theorem 5.9. Understanding that this particular coefficient will lead to the result is not obvious at all. Similarly, the coefficients $\mu n$ near $f_1$ and the coefficients $0$ near $f_2, \dots, f_n$ in the proof of Theorem 5.8 are also non-obvious. Each theorem has its non-trivial quadratic function construction.
>
> > The constructed function in the proof of Theorem 5.8 is a 2-dimensional quadratic with additive noise, which is not new at all.
>
> If you believe that our constructions are “not new at all,” we will be happy to reference the original paper that you think provides a similar construction.
>
> > The FL setting would indeed by of interest but as far as I can see the requirement for each worker to keep sending updates makes the studied methods inapplicable in that setting. If you're implying that asynchronous FL is impossible without this, it suggests that something is missing in the way the problem is formulated. A more reasonable setup would be if the workers are sampled and are not expected to communicate more than a single update (though potentially with many gradients).
>
> There are many FL settings and problems where all workers participate (e.g., [3,4] and many more). The full and partial participation settings are equally important for the community (we are not cherry-picking [3,4]; almost half of the works in FL and distributed learning do not consider the partial participation setting).
>
> We are not saying that asynchronous FL is impossible with partial participation. It is possible, but methods with partial participation are slower. Partial participation and our research are orthogonal questions. In this work, we want to bridge the gap between Malenia SGD and Rennala SGD under full participation of clients. Considering a version of Malenia SGD (let's call it PP-Malenia SGD) that supports partial participation would not bring us closer to the main question, since PP-Malenia SGD would be slower than Malenia SGD. We want to accelerate Malenia SGD, not slow it down. Even with full participation, it is not clear whether it is possible to accelerate Malenia SGD. Considering settings with partial participation might be the next research question.
>
> ---
>
> [1]: Y He, X Huang, K Yuan Unbiased compression saves communication in distributed optimization: When and how much? (NeurIPS 2023)
>
> [2]: Jiang R. et al. Provable Complexity Improvement of AdaGrad over SGD: Upper and Lower Bounds in Stochastic Non-Convex Optimization COLT 2025
>
> [3]: Richtarik P. et al. EF21: A New, Simpler, Theoretically Better, and Practically Faster Error Feedback NeurIPS 2021
>
> [4]: R Luo, SU Stich, S Horváth, M Takáč Revisiting LocalSGD and SCAFFOLD: Improved Rates and Missing Analysis AISTATS 2025
>
> [5]: Y. Arjevani, O. Shamir, N. Srebro, A tight convergence analysis for stochastic gradient descent with delayed updates (ALT 2020)

---

> > ### Comment · Reviewer_Mpme · 2025-11-17
> >
> > Low-dimensional quadratics with different scale of coordinates have been often used in the literature on lower bounds for gradient descent, e.g. some recent papers using it:
> > Woodworth et al. "Is Local SGD Better than Minibatch SGD?"
> > Cha et al. "Tighter Lower Bounds for Shuffling SGD: Random Permutations and Beyond"
> >
> > To be clear, I'm not asking the authors to cite these papers or any other, all I'm saying is that the construction itself is commonly used.

---

> > > ### Author Response · Authors · 2025-11-17
> > >
> > > Thank you for the response!
> > >
> > > > To be clear, I'm not asking the authors to cite these papers or any other, all I'm saying is that the construction itself is commonly used.
> > >
> > > It is also not clear how the popularity of quadratic functions or low-dimensional quadratic functions restricts us from using them. The fact that we use a low-dimensional quadratic function is clearly not a weakness of our work.
> > >
> > > One of the first cases where a quadratic function was used to prove a lower bound appears in the lectures “Introductory Lectures on Convex Programming. Volume I: Basic Course,” by Y. Nesterov (Lecture Notes, 1998). Does this mean that all subsequent works should avoid using quadratic functions? Of course, not. And there are many great works that use the particular construction from these lectures.
> > >
> > > We also use low-dimensional quadratic functions, and there is nothing wrong with this. There is also nothing wrong with the fact that they are low-dimensional; in fact, it is an advantage, since people generally aim to find constructions with the smallest possible dimension.
> > >
> > > Finally, our construction is unique. We chose particular quadratic functions with deliberately selected parameters.
> > >
> > > Thank you once again for participating in the discussion.

---

> > > > ### Comment · Reviewer_Mpme · 2025-11-17
> > > >
> > > > > It is also not clear how the popularity of quadratic functions or low-dimensional quadratic functions restricts us from using them. The fact that we use a low-dimensional quadratic function is clearly not a weakness of our work.
> > > >
> > > > I didn't say you shouldn't be using quadratic or low-dimensional functions to develop a lower bound, all I wanted to say is that this approach is standard. Since the studied method is weighted SGD, and lower bounds for SGD methods have been studied in prior work, the development of the lower bound is straightforward. The lower bound would have been much more insightful if it was applicable to a wider set of algorithms. For instance, the lower bound given in Nesterov's book, which you mentioned, applies to any first-order method, not just gradient descent.

---

> > > > > ### Author Response · Authors · 2025-11-17
> > > > >
> > > > > Thank you for the response.
> > > > >
> > > > > > I didn't say you shouldn't be using quadratic or low-dimensional functions to develop a lower bound, all I wanted to say is that this approach is standard. Since the studied method is weighted SGD, and lower bounds for SGD methods have been studied in prior work, the development of the lower bound is straightforward.
> > > > >
> > > > > As we previously said, it is difficult to respond to such a subjective comment as "the development of the lower bound is straightforward." We explained that the construction is non-trivial and new. If the reviewer would point to a paper with the construction
> > > > > $$
> > > > >       f_i(x, y) = \frac{\frac{\mu}{B_i}}{\frac{2}{n} \sum_{i=1}^n \frac{1}{B_i}} x^2 + \frac{L_{\max}}{2} y^2.
> > > > > $$
> > > > > from Theorem 5.9, we would agree with the reviewer. However, this construct is novel, and we don't know any paper with a similar lower bound. Papers Woodworth et al. "Is Local SGD Better than Minibatch SGD?" and Cha et al. "Tighter Lower Bounds for Shuffling SGD: Random Permutations and Beyond" analyze completely unrelated functions and setups.
> > > > >
> > > > > > The lower bound would have been much more insightful if it was applicable to a wider set of algorithms. For instance, the lower bound given in Nesterov's book, which you mentioned, applies to any first-order method, not just gradient descent.
> > > > >
> > > > > We acknowledge in the paper that our lower bounds do not capture the entire family of smooth convex or non-convex functions. However, the family of methods Weighted SGD is large enough to include two optimal methods, which justifies our research approach, and it is totally acceptable in the research community to focus on a subfamily of methods [1,2].
> > > > >
> > > > > [1]: Jiang R. et al. Provable Complexity Improvement of AdaGrad over SGD: Upper and Lower Bounds in Stochastic Non-Convex Optimization COLT 2025
> > > > >
> > > > > [2]: Y. Arjevani, O. Shamir, N. Srebro, A tight convergence analysis for stochastic gradient descent with delayed updates (ALT 2020)

---

### Official Review · Reviewer_m1iw · 2025-11-07

**Soundness:** 3
**Presentation:** 3
**Contribution:** 3
**Rating:** 6
**Confidence:** 4

**Summary:**

The paper addresses the difference in time complexity between heterogeneous and homogeneous settings. It aims to improve results in heterogeneous cases, narrowing the gap and aligning the complexity more closely with that of homogeneous problems. The authors introduce a unified framework called Loretta SGD for asynchronous algorithms, which generalizes Malenia SGD and Rennala SGD. They demonstrate that, under a strong interpolation regime and a local PL condition, convex functions with local smoothness can achieve results that match the performance observed in homogeneous settings.

**Strengths:**

The paper is solid, and the proofs and proposed theorems are logically sound and accurate. The convergence bound and complexity orders are carefully justified under appropriate assumptions, making the theoretical contributions reliable and well-supported.

The submission presents a derivation of the convergence results, aiming to find assumptions under which the performance of Rennala and Melania SGD coincide for homogenous and heterogeneous cases. They identify the PL condition, and strong interpolation would help, and thus derive the convergence in terms of E.

The paper is well-written, but its flow could be improved by adding a simpler discussion after each theorem because it leaves the reader confused about the main idea.

**Weaknesses:**

However, the submission has limited significance. Empirical results on quadratic optimization demonstrate a performance gap between Rennala SGD and Malenia SGD without interpolation; however, Rennala SGD does not converge to a minimum of the quadratic optimization problem. In the interpolation regime, the gap decreases, and Rennala SGD outperforms Malenia SGD, especially when devices have different computation times. Results on the CIFAR-10 dataset with Resnet-18 are also included. However, experiments with larger datasets and more data heterogeneity are missing. What happens when both variants of Loretta SGD are used on larger image and text datasets? What about the performance gap in those cases?

The submission is incremental compared to existing results. The paper examines how to reduce the gap between the convergence rates achieved by enhancing Malenia SGD's reliance on the arithmetic mean of computation times versus the harmonic mean. It shows that the assumptions of first- and second-order gradient similarity are insufficient to improve this dependence. They then apply the assumptions of weak interpolation and the PL-condition, but still do not achieve an improvement. Finally, the paper concludes that only under the strong interpolation regime, and when local functions satisfy the PL condition, does the time complexity improve.

**Questions:**

The reviewer identifies the paper from a previous submission, and last time, a query was also raised about the weak interpolation assumption. I am not convinced by the definition of weak interpolation in Assumption 4.1. Interpolation typically means that the model fits the data at each client, meaning the loss or gradient at each sample is zero. I believe the paper refers to the statement, “Formally, interpolation requires that the gradient with respect to each point converges to zero at the optimum, implying that if the function $f$ is minimized at $w*$ and thus $f(w*) = 0$, then for all functions $f_i$ we have $f_i(w*) =0$ in [1]. However, this definition aligns with the strong interpolation assumption presented in Assumption 5.6, rather than the weak interpolation assumption. There is no prior work that discusses a concept called weak interpolation; see the definition of interpolation in [2]. Therefore, I do not understand the need for this additional result with weak interpolation, especially since there is no clear intuition as to why this statement is valid. Interpolation indicates that all functions share common minimizers. If weak interpolation is assumed, then the reverse, namely, that if $x^*$ is a stationary point of each $f_i$, then it is also a stationary point of $f$, is trivially satisfied. Hence, weak interpolation automatically implies strong interpolation. Using other techniques, such as tuning $\gamma$ or incorporating correction terms, can’t we try to bridge the gap? Any discussion on that would make the conversation more well-rounded. Currently, the primary motivation is to determine how to make Rennala SGD work for heterogeneous cases by considering various theoretical assumptions. Does parameter tuning have any effect on this?

**REFERENCES**

[1] Sharan Vaswani, Aaron Mishkin, Issam Laradji, Mark Schmidt, Gauthier Gidel, and Simon Lacoste-Julien. Painless stochastic gradient: Interpolation, line-search, and convergence rates. NeurIPS 2019.

[2] Guillaume Garrigos and Robert Mansel Gower. “Handbook of Convergence Theorems for (Stochastic) Gradient Methods.” (2023).

---

> ### Author Response · Authors · 2025-11-16
>
> Thank you for the review. We now respond to the weaknesses.
>
> > However, the submission has limited significance. Empirical results on quadratic optimization demonstrate a performance gap between Rennala SGD and Malenia SGD without interpolation; however, Rennala SGD does not converge to a minimum of the quadratic optimization problem. In the interpolation regime, the gap decreases, and Rennala SGD outperforms Malenia SGD, especially when devices have different computation times. ... What about the performance gap in those cases?
>
> This is a theoretical work where we focus on the fundamental gap between the homogeneous and heterogeneous settings. Our main focus was to understand the theoretical nature of the methods and assumptions. Yet, we provided experimental results to test our theoretical results on both quadratic and modern classification tasks. In Figure 1, indeed, Rennala SGD does not converge, **which is expected** since Figure 1 considers the regime without interpolation. We acknowledge this in the discussion in Section I.1.
>
> > The submission is incremental compared to existing results. The paper examines how to reduce the gap between the convergence rates achieved by enhancing Malenia SGD's reliance on the arithmetic mean of computation times versus the harmonic mean. It shows that the assumptions of first- and second-order gradient similarity are insufficient to improve this dependence. They then apply the assumptions of weak interpolation and the PL-condition, but still do not achieve an improvement. Finally, the paper concludes that only under the strong interpolation regime, and when local functions satisfy the PL condition, does the time complexity improve.
>
> The final conclusion may indeed appear somewhat disappointing. However, our primary goal was to demonstrate that the considered assumptions are necessary. Simply stating them would not be convincing, so a central part of our paper is devoted to examining various alternative assumptions and showing that most of them fail to close the gap. *We believe that the significance of our contribution lies in this exploration process.*
>
> Questions:
> > The reviewer identifies the paper from a previous submission, and last time, a query was also raised about the weak interpolation assumption. I am not convinced by the definition of weak interpolation in Assumption 4.1. Interpolation typically means that the model fits the data at each client, meaning the loss or gradient at each sample is zero. ... **Hence, weak interpolation automatically implies strong interpolation.**
>
> Indeed, there might be a confusion. Please consider the updated PDF, where we clarified the assumptions. Weak interpolation does not imply strong interpolation, and [1] talks about the weak interpolation assumption. Let us consider a simple example.
>
> Take $x=(x_1,x_2)\in R^2$ and define two quadratic functions
> $$
> f_1(x)=\frac{1}{2}x_1^2,\qquad
> f_2(x)=\frac{1}{2}x_2^2,\qquad
> f(x)=\frac{1}{2}\bigl(f_1(x)+f_2(x)\bigr)=\frac{1}{4}(x_1^2+x_2^2).
> $$
> The function $f$ has the unique minimizer $x^*=(0,0)$, which is also a minimizer of each $f_i$, so weak
> interpolation holds. However, strong interpolation fails: for $\tilde{x}=(0,1)$ we have $f_1(\tilde{x})=0$
> (i.e., $\tilde{x}$ is a minimizer of $f_1$), while $f(\tilde{x})=\frac{1}{4}>0$, so $\tilde{x}$ is not a minimizer of $f$.
>
>
> > Using other techniques, such as tuning $\gamma$ or incorporating correction terms, can’t we try to bridge the gap? Any discussion on that would make the conversation more well-rounded.
>
> In this work, we investigate Weighted SGD with a fixed step size $\gamma$. Notice that the optimal methods, Rennala SGD and Malenia SGD, also use a fixed $\gamma$ and do not include any correction terms. Thus, it is reasonable to assume that Weighted SGD with a fixed step size $\gamma$ is a general enough family for investigation. Even in this case, we believe that the results and theorems are surprising and non-trivial. We conjecture that even with tuning, our conclusions would still hold. Extending this to other families of algorithms is an important direction for future work.
>
> > Currently, the primary motivation is to determine how to make Rennala SGD work for heterogeneous cases by considering various theoretical assumptions.
>
> The primary motivation is to analyze a more general family of methods, Weighted SGD, not only Rennala SGD. Rennala SGD is a special case of Weighted SGD. We focus on a broad family of methods.
>
> ---
>
> **Thank you for your comments. Please consider the improved definitions of the assumptions in the updated PDF and the example, where we have hopefully resolved all confusion.**
>
> [1] Sharan Vaswani, Aaron Mishkin, Issam Laradji, Mark Schmidt, Gauthier Gidel, and Simon Lacoste-Julien. Painless stochastic gradient: Interpolation, line-search, and convergence rates. NeurIPS 2019.
>
> [2] Guillaume Garrigos and Robert Mansel Gower. “Handbook of Convergence Theorems for (Stochastic) Gradient Methods.” (2023).

---

### Author Response · Authors · 2025-12-03
**Public Summary Comment by Authors**

Dear AC and Reviewers,

Thank you for participating in our submission! We would like to briefly summarize our responses to the reviewers’ weaknesses and questions, as well as our main contributions. Let us briefly outline the main concerns and clarify our responses:

---

Reviewers **m1iw** and **mKAf** gave us positive scores and explained in their comments that our work is solid, well written, and provides nice and insightful takeaways. They also pointed out several weaknesses and questions, which we believe are addressed in our rebuttal: either we clarify questions, or we acknowledge minor inaccuracies or missing references, which we have already added to the updated PDF.

---

Reviewer **SYHQ** also agrees that our paper "addresses a well-defined question" and "presents a sound and convincing argument" to obtain a clear final result. The reviewer raised several weaknesses and questions. We believe that the concerns raised, for example "*I am also not fully convinced by the framing of the paper. It would be good if the authors could provide more intuition about what they are trying to do and why?*" or "*Can you elaborate on what is fundamental with the harmonic mean compute-time dependence in the heterogeneous setting?*", are points that we clarified in our rebuttal. All these questions are fully addressed in our comments.

---

Reviewer **Mpme** raised the following concerns:

> **(A):** "I'm also concerned about how the results for the similary are presented."

The presentation concern **(A)** is minor and already fixed in the updated paper.

> **(B):** "More generally, what kind of setting are the authors motivated by? It doesn't appear that we have data heterogeneity in distributed computation on clusters as we can simply duplicate the data."

A heterogeneous setting is standard in FL due to privacy constraints. We clarified this motivation in the rebuttal and explained why both homogeneous and heterogeneous settings are relevant.

> **(C):** "I also feel the topic of the paper is drifting from the practical methods. Malenia SGD requires each worker to submit at least one gradient, which is not something we would use in practice when a node failure is possible."

The reviewer asks why we do not consider the partial participation setting. We explain that there are many FL settings and problems where all workers participate. Both full and partial participation settings are widely studied, and many FL problems operate in the full participation regime. We clarified this in the rebuttal.

> **(D):**  "The studided methods were proposed and analyzed in prior literature. The non-convergence result is mostly trivial. Moreover, the results for first-order and second-order similarity are only applicable to Weighted SGD, which in itself is not a particularly interesting method."

Our additional discussion was around **(D)**, where the reviewer says that “the construction itself is rather straightforward.” We explained that, to the best of our knowledge, the construction we use does not appear in prior literature. For instance, the construction $$ f_i(x, y) = \frac{\frac{\mu}{B_i}}{\frac{2}{n} \sum_{i=1}^n \frac{1}{B_i}} x^2 + \frac{L_{\max}}{2} y^2. $$ from Theorem 5.9 with its coefficients is nonstandard and never appeared previously.

The last concern in **(D)** was about the fact that we consider the Weighted SGD family of methods. We consider Weighted SGD since it is a broad family of methods that naturally generalizes both Rennala SGD and Malenia SGD, which are specific instances of Weighted SGD with particular choices of weights. There are many important works that analyze families of methods, with significant implications for the optimization community (e.g., [1] considers only Asynchronous SGD, [2] focuses only on Local SGD). In the paper, we acknowledge that we consider the Weighted SGD family rather than all possible optimization methods. However, even within this scope, the overall contribution was viewed positively by Reviewers **m1iw** and **mKAf**.

---

Thank you once again for your time!

Authors

---

[1]: Y. Arjevani et al., A tight convergence analysis for stochastic gradient descent with delayed updates (ALT 2020)

[2]: MR. Glasgow et al., Sharp bounds for federated averaging (local sgd) and continuous perspective (AISTATS 2022)

---

### Meta-Review · Area_Chair_ebn1 · 2026-01-06

**Summary:**

This paper examines the time complexity gap between homogeneous and heterogeneous asynchronous optimization, and inquires about the assumptions required to recover the more favorable worker-time dependence characteristic of the homogeneous case. The technical approach involves analyzing a Weighted SGD family that subsumes the two previously studied schemes, which are optimal in the homogeneous and heterogeneous settings, respectively. The paper presents a structured sequence of negative results, demonstrating that several natural assumptions (including gradient and Hessian similarity, as well as a weak interpolation condition) do not close the gap within this family. It then presents a positive result, showing that strong interpolation, combined with a local PL condition, is sufficient to recover the desired dependence.

The results appear correct and the narrative is coherent. The main reviewer concern is significance: the strongest impossibility statements apply only to the Weighted SGD class, and the paper does not provide a compelling justification that this class captures the full range of reasonable asynchronous methods one might try.

**Reviewer Concerns:**

The positive reviewers emphasize clarity, technical soundness, and the value of systematically testing which common assumptions fail to close the gap. The more critical reviewers focus on scope and positioning. They argue that the paper’s broad motivation is not fully matched by its restricted focus on the Weighted SGD family, and that the paper does not justify why this family should be the right boundary for the main claims. As a result, it remains plausible that other design choices, such as correction terms, adaptivity, variance reduction, or different asynchronous protocols, could lead to different trade-offs.

The discussion and revisions enhance the presentation and refine the claims, including the clarification of interpolation definitions with explicit examples. One reviewer questions whether the lower-bound constructions are genuinely novel (although the specific parameter choices are new and tailored to this setting), though this is a minor point.

**Reviewer Scores:**

Reviewer m1iw and Reviewer mKAf would likely stay at 6. Reviewers Mpme and SYHQ, who express high confidence, would likely stay at 4, since their main concern about the narrow scope is not resolved.

---

### Decision · Program_Chairs · 2026-01-26

Reject